# Observation of a robust and active catalyst for hydrogen evolution under high current densities

Yudi Zhang[1,2], Kathryn E. Arpino[3], Qun Yang[3], Naoki Kikugawa[4], Dmitry A. Sokolov[3], Clifford W. Hicks[3], Jian Liu[2,5] ✉, Claudia Felser[3] ✉ & Guowei Li[1,2] ✉

Despite the fruitful achievements in the development of hydrogen production catalysts with record-breaking performances, there is still a lack of durable catalysts that could work under large current densities (>1000 mA cm$^{-2}$). Here, we investigated the catalytic behaviors of $Sr_2RuO_4$ bulk single crystals. This crystal has demonstrated remarkable activities under the current density of 1000 mA cm$^{-2}$, which require overpotentials of 182 and 278 mV in 0.5 M $H_2SO_4$ and 1 M KOH electrolytes, respectively. These materials are stable for 56 days of continuous testing at a high current density of above 1000 mA cm$^{-2}$ and then under operating temperatures of 70 °C. The in-situ formation of ferromagnetic Ru clusters at the crystal surface is observed, endowing the single-crystal catalyst with low charge transfer resistance and high wettability for rapid gas bubble removal. These experiments exemplify the potential of designing HER catalysts that work under industrial-scale current density.

For catalysis reactions such as hydrogen evolution, the use of catalysts is essential to overcome the activation barriers, which are significantly larger than the theoretical minimum of 1.23 V[1–5]. Though considerable progress has been made in the search for high-performance catalysts, and some of them even outperform the state-of-the-art noble catalysts[6–10], it remains a challenge to run the reaction efficiently and economically. Thus, the identification of durable catalysts that could afford the industrial scale current density (>500 mA cm$^{-2}$) is crucial for the forthcoming hydrogen economy[11,12].

The working mechanisms under low and high current densities are fundamentally different for the same catalyst[13]. In addition to the empirical rule known as the Sabatier principle, the interfacial charge transfer resistance from the bulk phase to the surface, reaction intermediates coverage, catalysts mechanical stability, and hydrogen bubble release kinetics should be also taken into consideration in dealing with high current density mechanisms[14–17]. At first, the rate-determining step in the cases of low current densities and high current densities is different. Most works report their Tafel slope values at relatively low current densities below 100 mA cm$^{-2}$, which are generally around 30 mV dec$^{-1}$ or even lower[18]. However, this value can be increased quickly to above 120 mV dec$^{-1}$ with the increase of overpotential and current densities, even including the state-of-the-art Pt catalysts, suggesting the vital role of the diffusion of protons and the adsorption of hydrogen issues[19]. Secondly, hydrogen bubble release kinetics should be given adequate attention. Large bubble size and high level of bubble coverage at the crystal surface will cover and kill the active sites, induce local strain, and are detrimental to intrinsic activity and stability[20,21]. Conductivity is another important issue that needs to be taken into consideration. The phenomenology can be well explained by the studies on layered transition metal dichalcogenides (TMDs)[22,23]. Electron hopping across the layer is unfavored because of the large interlayer potential barriers. This will lead to sluggish

[1]CAS Key Laboratory of Magnetic Materials and Devices, and Zhejiang Province Key Laboratory of Magnetic Materials and Application Technology, Ningbo Institute of Materials Technology and Engineering, Chinese Academy of Sciences, Ningbo 315201, China. [2]University of Chinese Academy of Sciences, 19A Yuquan Rd, Shijingshan District, Beijing 100049, China. [3]Max Planck Institute for Chemical Physics of Solids, Nöthnitzer Strasse 40, 01187, Dresden, Germany. [4]National Institute for Materials Science (NIMS), Tsukuba 305-0003, Japan. [5]Center for Advanced Solidification Technology, School of Materials Science and Engineering, Shanghai University, Shanghai 200444, China. ✉e-mail: liujian@shu.edu.cn; claudia.felser@cpfs.mpg.de; liguowei@nimte.ac.cn

electron transfer through the bulk phase and the subsequent injection into the surface adsorbates. Most importantly, the low conductivity will lead to misleading conclusions because of the excessive Ohmic drop (iR) correction, which is meaningless for particle applications[24,25]. For instance, an impressive overpotential of 382 mV is reported at the current density of 1000 mA cm$^{-2}$ for the modified $MoS_2$ catalysts, however, the actually applied potential is 1400 mV before iR correction[26].

Transition metal oxides with layered structures have emerged as one of the major testing grounds for experimental and theoretical investigation of electronic structures, surface reconstructions, and the associated electrochemical properties[27–30]. Although, in most cases, their thermodynamically stable in-plain surfaces are inert towards catalysis, they can be modified or reconstructed to achieve high efficiencies. Benefiting from the large exposed surface areas, active metal catalysts can be loaded onto these surfaces with techniques such as drop-casting, atomic layer deposition, chemical vapor deposition, and exsolution[31]. However, aggregation and deactivation issues are still challenges under harsh experimental conditions such as high current densities and potentials. Recently, in situ exsolution has been proven to be a promising method to load highly active metal catalysts[32,33]. In this method, the exsolved metal nanostructures are socketed into the bulk oxide phases, leading to enhanced cohesion between the produced metal catalysts and the supports. This guarantee the well-dispersion and good stability of the modified catalysts, favoring the electrochemical performance because of the strong interaction and the accelerated electron transfer kinetics[34,35]. As the closest 3D analog to 2D materials, layered oxide perovskite $Sr_2RuO_4$ (SRO) has attracted intense interest because of the d orbital electronic correlation-derived exotic properties such as superconductivity, surface magnetism, and good metallicity[30,36,37]. These properties, in turn, will influence its catalysis applications. Few works have been carried out to reveal SRO's

catalytic properties, but mostly with polycrystalline samples and under a low current density range[38–40].

In this work, we examine the catalytic properties of bulk SRO single crystals in millimeter size. As HER catalysts, the single crystals exhibit remarkable activities and stabilities under the high current density above 1000 mA cm$^{-2}$. The TOF value is determined to be 121 s$^{-1}$ at 100 mV, which sets it as one of the state-of-the-art catalysts thus far. We find the in situ formation of ferromagnetic Ru clusters at the surface of the bulk single crystal after activation. In combination with DFT calculations, we confirm that the charge redistribution at the interface between Ru clusters and the bulk SRO, the excellent bulk conductivities, and the optimized wettability are responsible for the observed high performance[41]. Our studies shed new light on the design of durable catalysts that could withstand harsh conditions of industrial-scale hydrogen production.

## Results and discussion
### Synthesis and crystal structures of SRO single crystals
SRO adopts a body-centered tetragonal perovskite structure with the space group of I4/mmm. The SrO rocksalt and $SrRuO_3$ perovskite sandwiched layers are alternatively arranged along the c direction, making the cleaving and exposing of the ab plane energetically preferable (Fig. 1a). Single crystals of SRO were then grown by the floating-zone method as described elsewhere, with their phases and physical properties have been studied extensively by our group and collaborators[42–44]. A piece of plate-like crystal (~3 mm × 2 mm × 0.2 mm) was cut from the SRO single crystal rods and used for this investigation (inset Fig. 1b), with its crystal structure has been confirmed by the Laue method of X-ray diffraction (Fig. 1b). Scanning electron microscope (SEM) images recorded with secondary and backscattered electrons suggest the homogeneity of the chemical composition of the surface of the

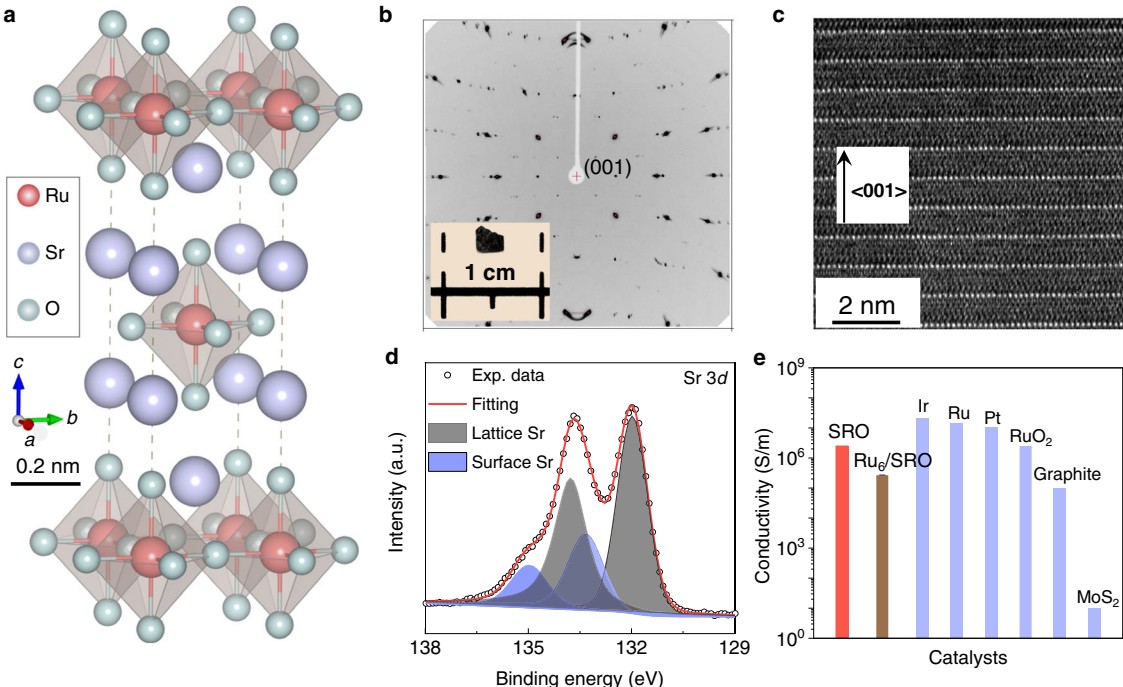

**Fig. 1 | Crystal structure of fresh $Sr_2RuO_4$ single crystals. a** Three-dimensional crystal structure of layered SRO crystal. The light purple, pink, and gray balls represent Sr, Ru, and O atoms, respectively. **b** Laue diffraction pattern of the SRO crystal recording along the [100] direction. A photograph of the exfoliated SRO single crystal by exposing the (001) surface. **c** TEM image of the fresh SRO crystal recorded on a FIB laminar. The bright atoms represent the alternatively arranged Sr element along the c direction. **d** XPS spectrum of the SRO catalysts taken at Sr 3p. The peak can be fitted well with lattice and surface Sr components. **e** Comparison of the room temperature conductivities between SRO single crystal and reported state-of-the-art HER catalysts. The error bar in conductivity of the activated SRO crystal is based on three independent measurements.

cleaved crystal (Fig. S1a, b). A lamella perpendicular to the *ab* plane is fabricated by the focused ion beam technique for transmission electron microscopy (TEM) observation, where we can see the atomically resolved lattices that are stacked along the *c* direction (Fig. 1c). Polarized Raman spectra of the SRO crystal further confirmed the symmetry of the surface crystal structures. Two of the $A_{1g}$ modes are Raman active and can be attributed to the Sr-Sr and apex O-O stretching vibrations along the *z*-direction (Fig. S2)[45]. It should be noted that a minority contribution of $Sr_3Ru_2O_7$ interlayer inside the bulk of this sample was observed. However, this does not change the surface electric and crystal structures, as well as the high conductivity of SRO[46].

The surface electronic structures of SRO single crystal are investigated by X-ray photoelectron spectroscopy (XPS). The survey spectrum demonstrates the existence of Sr, Ru, and O elements besides the common C contamination (Fig. S3). By using the Shirley background subtraction and mixed Gaussian–Lorentzian peak shapes, excellent fitting of the Sr 3*d* region with two doublets is obtained (Fig. 1d). The doublets at the lower binding energy (Sr $3d_{5/2}$ = 132.0 eV) correspond to the "lattice" component of the SRO crystal in the near-surface region, while the one with higher binding energy ($3d_{5/2}$ = 133.3 eV) can be indexed to the "surface" component that associated with the perovskite surface termination, which is consistent with previous reports on the Sr-based perovskites[47,48]. Particular attention should be paid to understanding the Ru spectrum because of the strong overlapping between the Ru 3*d* doublets and the C 1*s* signal. But satisfactory fitting still can be obtained with two sets of doublets (Fig. S4). The peak with lower binding energy (Ru $3d_{5/2}$ = 280.8 eV) corresponds to the screened states of Ru 3*d* orbitals in SRO. The broader doublets sitting at the higher binding energy (Ru $3d_{5/2}$ = 282.2 eV) are associated with the unscreened core-hole state. The results are fully consistent with previous theoretical and experimental observations on SRO and other layered perovskite ruthenates because of the charge transfer via the Ru 4*d*-O 2*p* bonding[49]. The observation of the unscreened satellite peak suggests a strong electron correlation in SRO, but is relatively weak concerning its siblings, such as $SrRuO_3$ and $Sr_3Ru_2O_7$[37]. This well explains the high conductivity in SRO on the order of $10^6$ S/m, which is comparable to the most conductive metals such as Ir, Ru, and Pt, and much higher than graphite and most metal sulfides (Fig. 1e). Additionally, we also measurement the conductivity of the SRO crystal after long-time stability test. Although the conductivity is decreased to about $3.0 \times 10^5$ S/m, it still makes it one of the most conductive catalysts reported so far.

## HER catalytic performances

Having established the crystal and surface electronic structures, we now turn to the HER catalytic investigation. We attached the bulk single crystal of SRO with Cu wire with silver paint and used it as the working electrode (more details can be seen in the experimental section). Before the assessment of the catalytic performance, we test the HER activities of Cu wire and silver paint to make sure that their contributions can be neglected (Fig. S5). Polarization curves suggest that the required overpotential to reach the current density of 10 mA cm$^{-2}$ for the SRO single crystal is 18 and 28 mV in 1 M KOH and 0.5 M $H_2SO_4$ electrolyte, respectively (Fig. 2a). These values are comparable with the state-of-the-art noble metal catalysts such as Pt/C (17 mV), Pt single-atom catalysts (26 mV)[50], and Ru metal on two-dimensional carbon (22 mV)[51], although the specific surface area of the SRO single crystal is much smaller than most nanostructured catalysts. Tafel slopes were determined to be 22 and 29 mV dec$^{-1}$, which is close to that of the Pt/C (30 mV dec$^{-1}$), implying fast HER kinetics (Fig. 2b). The low Tafel slope in alkaline electrolytes suggests that the rate-determining step should be the H desorption process, rather than the water dissociation (Volmer step)[50,52]. Turnover frequency (TOF) per real active site of the SRO catalyst is obtained by measuring the electrochemical

surface areas (ECSAs) (Fig. S6a, b) (note: the TOF calculations are based on the real structure after activation, rather than the pristine SRO, more information can be seen in Figs. 3, 4). The ECSA of the activated catalyst is determined to be 250 m². At the overpotential of 100 mV, the TOF value is calculated to be 121 s$^{-1}$ for the SRO single crystal catalyst, which is higher than most of the recently reported state-of-the-art catalysts (Fig. 2c and Table S1)[1,14,53–57]. Further considering the fact that the ECSA of the SRO catalyst is much smaller than the reported noble metal-based catalysts (Fig. S7), we can make the conclusion that the activated SRO belongs to one of the best HER catalysts at low current densities (<100 mA cm$^{-2}$).

The bottleneck problems for large-scale industrial applications are insufficient efficiency and stability at large current densities above 500 mA cm$^{-2}$[11]. Here we tested the HER activities of SRO crystal under critical conditions. To reach the current density of 1000 mA cm$^{-2}$, the SRO catalyst requires overpotentials of 182 and 278 mV after iR correction in $H_2SO_4$ and KOH, respectively (Fig. 2d and Figs. S8, 9). It is recently proposed that for practical water electrolysis, iR correction is less meaningless as they are an integral part of total overpotentials[25]. Corresponding polarization curves without iR corrections suggest that the overpotentials are increased slightly to 272 and 354 mV in $H_2SO_4$ and KOH electrolytes, respectively. The increases in overpotentials before and after iR correction (~90 mV) are much smaller than other high-performance catalysts such as NiMoN (546 mV)[58], MoNi$_4$ (632 mV)[59], and Se/Co modified $MoS_2$ (1018 mV)[26]. By comparison of the required overpotentials to reach the current density of 1000 mA cm$^{-2}$ with the state-of-the-art catalysts (Fig. 2e and Table S2), one can expect the potential of using SRO for high-current-density water electrolysis technologies. The quick and stable response of current to the change of applied potentials from the multi-step chronopotentiometry measurement suggests the high robustness and high mechanical stability of single-crystal electrodes even under a high current density of 9000 mA cm$^{-2}$ (Fig. S10). Mass activities is calculated to check the potential application of SRO/Ru6 catalyst for practical hydrogen production. We assume that the HER activities are completely derived from the Ru clusters at the crystal surface. The mass activities of SRO/Ru is determined to be 16.6 A mg$^{-1}$ at an overpotential of 50 mV, making it one of the best HER catalysts among the noble metal-based compounds[60–62], although the value is still lower than some Ru-based nanostructures (Fig. 2f and Table S3)[63,64].

The durability of catalysts under long-time operations and/or under harsh conditions are another important criteria to assess the performance. At a fixed overpotential of ~425 mV (without iR correction), the SRO catalyst delivered a current density of ~2000 mA cm$^{-2}$ under acidic conditions and maintained good stability during a test of 5 days (Fig. 2g upper panel). Stability in the alkaline electrolyte was also examined because of its highly competitive and commercial applications. At the room temperature of 25 °C in 1 M KOH electrolyte, the SRO catalysts exhibited impressive stability for a measurement of 35 days at the current density of 1000 cm$^{-2}$ (Fig. 2g lower panel). The high stability can be further confirmed by the comparison of LSV curves before and after the long-time test, with negligible loss of activities (Fig. S11). We then increased the electrochemical system to the working temperature (70 °C) of industrial electrolyzers. At a fixed overpotential of 354 mV (corresponding to 272 mV after iR correction), the current density is increased to above 1300 mA cm$^{-2}$ and exhibited superior stability during a test of 21 days (Fig. 2g lower panel). We further investigated structure changes after catalysis at a high testing temperature of 70 °C. SEM images coupled with elemental mapping suggested a similar activation mechanism in comparison with the room temperature test (Fig. S12a–e). The formation of the Ru metal layer was finally proved by the strong signal of the Ru metal peak from XPS measurements (Fig. S12f). In addition, we did not observe significant changes in HER reaction kinetics and hydrogen desorption behavior from the corresponding Tafel and bubble release kinetic

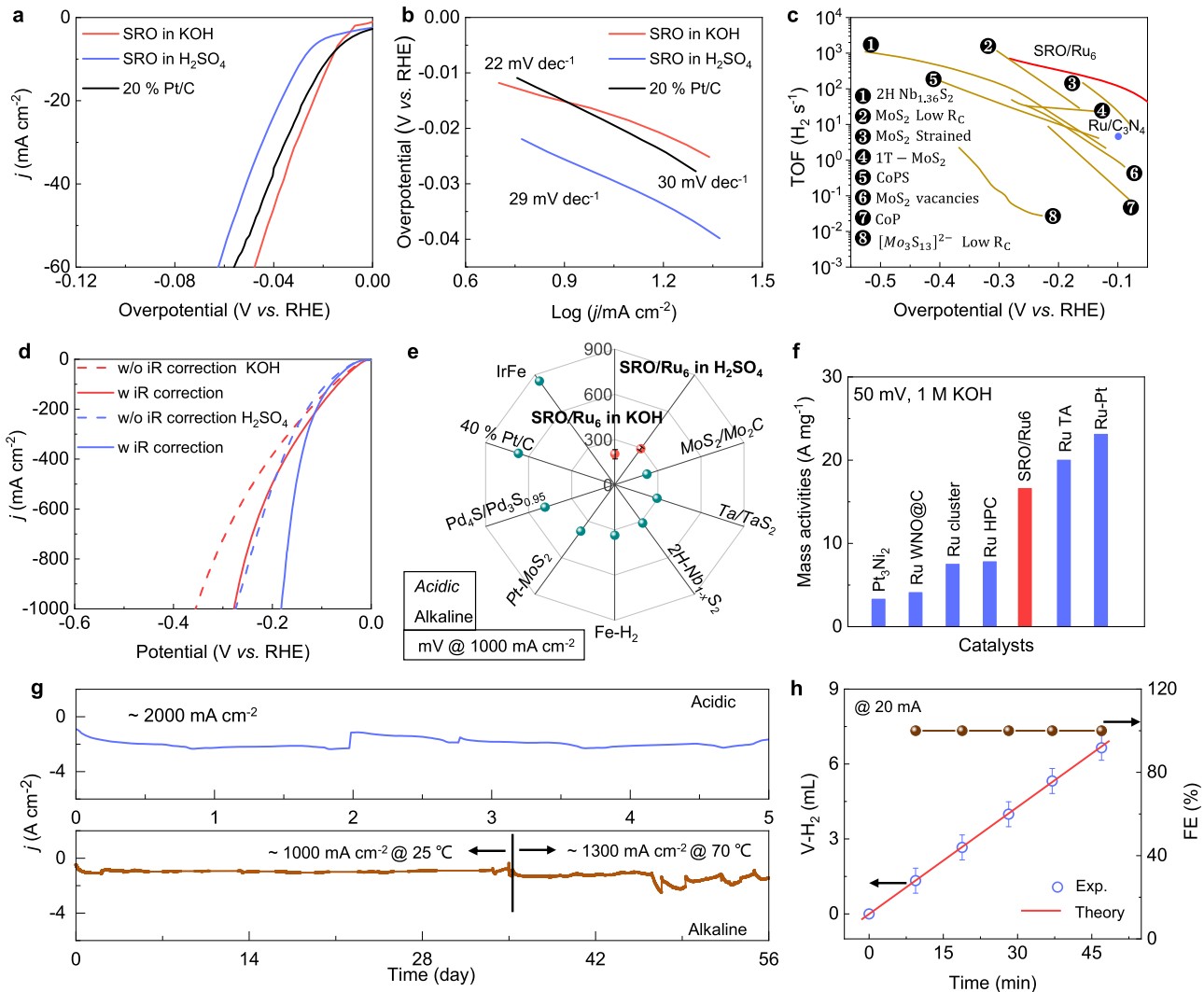

**Fig. 2 | HER performance of SRO single crystal catalysts. a** HER polarization curves and **b** the corresponding Tafel slopes of SRO catalyst in 0.5 M H$_2$SO$_4$ and 1 M KOH electrolytes. The LSV curve of Pt/C in 0.5 M H$_2$SO$_4$ is shown as a comparison. **c** Comparison of TOF values of the SRO catalysts with recently reported state-of-the-art catalysts. **d** LSV curves of the SRO catalyst plotted in a larger overpotential range with (w) and without (w/o) considering the ohmic drop. **e** Comparison of required overpotentials to reach the current density of 1000 mA cm$^{-2}$ between the activated SRO catalyst and recently reported advanced catalysts. Scale bar of the SRO catalyst are based on parallel LSV measurements. **f** Comparison of mass activities between activated SRO crystal and noble metal-based HER catalysts. **g** Long-term stability test of the SRO catalyst in acidic and alkaline conditions at room temperature and 70 °C (1 M KOH). **h** Faradaic efficiency of SRO for the theoretically calculated and experimentally measured H$_2$ at a current density of 1000 mA cm$^{-2}$. Error bars are based on the measuring accuracy of the graduated cylinder at 20 °C.

analysis (Fig. S13). To check the Faraday efficiency (FE) of the catalyst and make sure that the observed current is indeed from the hydrogen production, we collected the produced hydrogen gas at a current density of 1000 cm$^{-2}$ (Fig. 2h). The FE is determined to be close to 100% by comparing the theoretically produced hydrogen and collected gases in reality. Now we can safely claim that the SRO single crystals represent one of the best HER catalysts thus far. Finally, a completer electrolyzer was assembled by using the activated SRO/Ru catalyst as the cathode and the commercial 10% Ir/C powder (Aladdin) as the anode. To reach the current density of 1000 mA cm$^{-2}$, the SRO /Ru ‖ Ir/ C pair requires a potential of 1.87 V when considering iR drop (Fig. S14a). It also exhibits excellent electrochemical stabilities during a long-time test of 24 h under such a high current density(Fig. 14b).

## Crystal structure and activities evolution during HER

The large size of the bulk single crystal serves as a good platform to investigate the evolution of crystal structures during the catalytic reaction, which is the basis for the precise understanding of catalytic

mechanisms. We observed a fast activation of the SRO single crystal from the cyclic voltammetry measurements (Fig. 3a). The current density is increased from 100 mA cm$^{-2}$ for the first cycle to around 400 mA cm$^{-2}$ in the 30th cycle with a CV scan speed of 50 mV/s. The inset of Fig. 3a displayed the enlarged view of the stability test curves of Fig. 2g lower panel, from where we can see that the activation process takes about 15 h. The accelerated HER kinetics can be seen from the Nyquist representations of the impedance spectrum (EIS) before and after activations at the same overpotential (Note: the definition of before activation doesn't mean there is no change in SRO. The activation process starts quickly in several tens of seconds). The impedance spectra can be fitted well with the two-semi arcs model proposed by ref. 65 (Fig. 3b and Fig. S15). Electrical equivalent circuits with two-time constants were used to understand the spectra, with one representing the interfacial charge transfer kinetics connected to the electrosorption process and the other one for the resistivity as a result of the dielectric interlayer. In combination with the corresponding EIS Bode plots (Fig. 3c), we can attribute the time constant at low

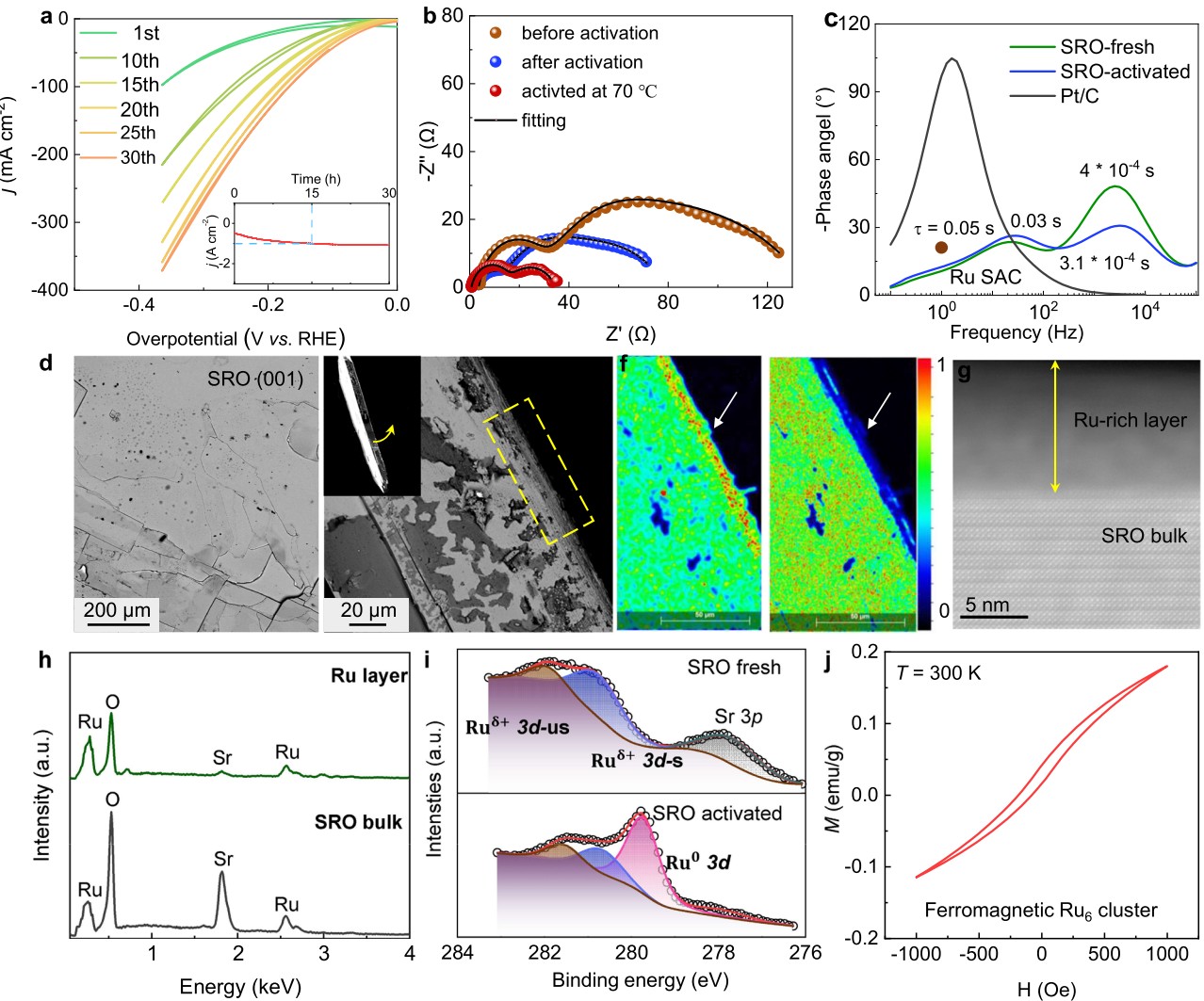

**Fig. 3 | Structure of the SRO crystals after surface reconstruction. a** Recorded CV curves in the first 30 cycles with the fresh SRO crystal. **b** Nyquist plots and **c** the corresponding Bode plots of SRO catalyst before and after HER testing. The used equivalent circuit to model the FRA spectra can be found in the supporting information. Ru SAC represents Ru single-atom catalyst. **d** SEM image of the SRO crystal surface after activation. **e** SEM image of the side view. The area marked with a yellow rectangle indicates the SRO surface that comes in contact with the electrolyte. **f** Elemental mapping suggests the accumulation of Ru and the depletion of Sr element in the surface area. **g** TEM image was recorded on a FIB laminar after activation, indicating the existence of an amorphous Ru-rich layer. **h** EDS spectra of the reconstructed catalysts in the surface and bulk phase. **i** Comparison of the Ru 3d XPS spectra before and after activation. *s* and *us* represent the screened and unscreened Ru 3d peak. **j** Hysteresis loops of the activated SRO crystal, indicate the existence of a ferromagnetic phase.

frequency (10–100 Hz) to the kinetics of the interfacial charge transfer reaction. The charge relaxation time is decreased from 0.05 to 0.03 s after activation, suggesting the improved HER transfer kinetics. Interestingly, we found that the charge relaxation time of our sample is even shorter than the state-of-the-art commercial Pt/C catalysts (0.62 s) and Ru single-atom catalysts (0.08 s)[66]. It is interesting to see a rapid relaxation process ($10^{-4}$ s) in the high-frequency ($10^3$–$10^4$ Hz) regime, which is two orders of magnitude lower than the interfacial charge transfer process. This is rarely reported before and is generally limited to some iron-based or strongly surface reconstructed catalysts[67]. Although debate still exists for the origin of such a short timescale relaxation, more and more evidence suggests that the formation of a dielectric interlayer between the bulk substrate (SRO) and highly conductive outer layer (Ru clusters) is expected[65].

As expected, obvious surface reconstruction is observed by comparing the SEM images of the fresh SRO crystal surface (Fig. S16) and the one after activation (Fig. 3d and Fig. S17) (Note: The crystal in Fig. S16 is a freshly exfoliated crystal). The roughness of the surface increased significantly with the observation of numerous micro-scale

islands, which is the same as the observation of the metal delafossite oxides bulk single crystals[28]. More details can be seen from the SEM images taken from the cross-section of the crystal (Fig. 3e and Fig. S18a). The top layers that are in contact with electrolytes are strongly distorted and stacked loosely. Elemental mapping on the cross-section suggests the accumulation of Ru in the top region and the loss of Sr (Fig. 3e right and Fig. S18b). TEM laminar cutting from activated single-crystal indicates the in situ formation of an amorphous layer at the surface bulk crystalline SRO (Fig. 3g and Fig. S19). EDS spectra recording from the reconstructed layer and the internal bulk SRO further confirmed the absence of the Sr element in the topmost layers (Fig. 3h and Fig. S20). After the long-term stability test at 70 °C, the concentration of Sr and Ru elements in the electrolyte is determined to be 0.66 ppm and 0.02 ppm by inductively coupled plasma measurements, which explains the preferred leaching of Sr.

The component of the reconstructed layer is determined to be amorphous Ru clusters, rather than other possibilities such as $RuO_2$, and bulk Ru. The hypothesis is well proved by comparing the XPS spectra before and after activation (Fig. 3i). From the fitted Ru $3d_{5/2}$

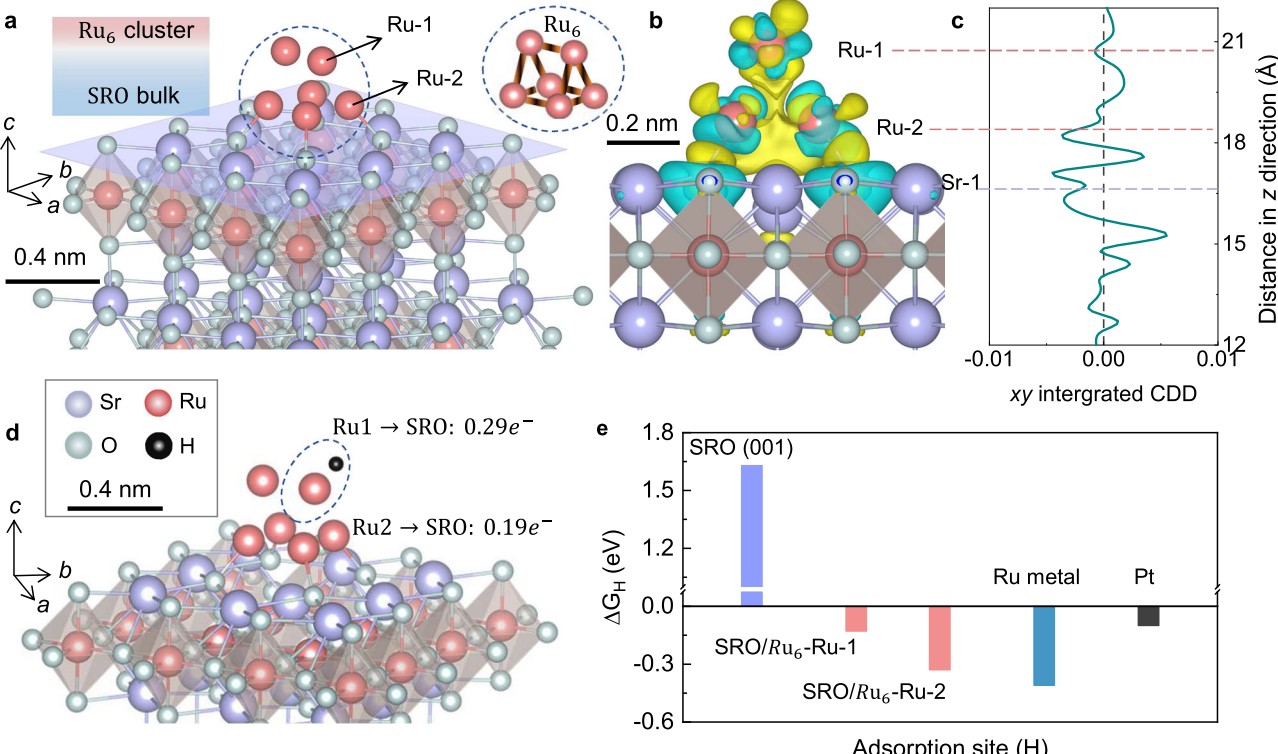

**Fig. 4 | Mechanism of the HER process at the surface of reconstructed SRO.**
**a** Optimized adsorption geometry for the Ru$_6$ cluster at the (001) surface of SRO. Ru1 and Ru2 represent the Ru site that binding with the SRO and above the surface. The Ru cluster with $D_{3h}$ geometry is given for reference. **b** Charge-density difference and **c** the corresponding plane-averaged charge-density difference (right) along the z-direction normal to the Ru$_6$/SRO interface. **d** Optimized hydrogen adsorption of H atom at the surface of Ru$_6$/SRO. Charge transfer numbers are different for the Ru1 and Ru2 sites. **e** Comparison of $\triangle G_{H^·}$ values of the fresh SRO, activated Ru$_6$/SRO catalyst, and state-of-the-art Pt catalyst.

spectrum, we can see that there are no changes for the doublets belonging to the screen and unscreened peaks of Ru 3d in SRO. However, a strong peak located at the binding energy 279.8 eV is observed, which has been attributed to Ru$^0$ metal. Accordingly, we found a significant decrease of Sr bound in bulk SRO lattice components from the Sr 3d spectrum (Fig. S21). More information can be obtained from the magnetization measurements. It is not strange that the activated bulk single-crystal exhibited paramagnetic properties at room temperature because the bulk SRO crystal itself is paramagnetic down to 1.5 K (Fig. S22)[68]. Interestingly, we can see an obvious magnetic hysteresis at low magnetic fields, indicating the existence of a ferromagnetic phase with Curie temperature ($T_c$) above room temperature (Fig. 3j and Fig. S23). We checked the magnetic properties of all the possible phases that could be in situ produced based on the bulk SRO crystal, including Ru cluster, Ru bulk metal (Note: bulk means in microsize or bigger and beyond the nanosize effect), RuO$_2$, SrO, and Sr. It is found that only Ru clusters exhibit ferromagnetic behaviors with $T_c$ above room temperature[69].

**Theoretical understanding of the enhanced HER mechanisms**
Having established the fact that the reconstructed Ru clusters on the SRO crystal surface are the real catalysts and responsible for the observed high performance, we now turn to the question of why? The definite answer can be obtained by exploring the Gibbs free energy of hydrogen adsorption ($\triangle G_{H^·}$). We first studied the hydrogen adsorption behaviors on the (001) surface of pristine SRO crystal. [RuO6] octahedral unit is exposed because of the relatively weak van der Waals force between the layers. As expected, very positive $\triangle G_{H^·}$ values of 1.63, 1.64, and 1.76 eV are obtained for the hollow sites, O-top sites, and Sr-top sites, respectively (Fig. S24 and Table S4), suggesting the unfavorable hydrogen adsorption and sluggish HER thermodynamics.

We then investigated the role of surface reconstructions in the presence of Ru clusters. For this purpose, the configuration of Ru clusters should be carefully chosen. We reviewed the crystal and electronic structures of Ru clusters with different coordination numbers, and finally selected the Ru$_6$ configuration because of its room temperature ferromagnetism and thermodynamically favorable formation[69–71]. The hypothesis is finally supported by the comparison of $\triangle G_{H^·}$ values between different Ru cluster/SRO geometries (Fig. S25 and Table S5). The stability of two possible geometrics of Ru$_6$ clusters ($D_{3h}$ and $D_{2h}$) on SR O (001) surface are explored, where we find that only the $D_{3h}$ geometry is stable, which is consistent with previous studies[72] (Fig. 4a, upper right). In addition, we tried four different adsorption geometries of $D_{3h}$ on SRO (001), and found the most favorable one is the bonding of four Ru atoms on the SRO surface (with binding energy as −6.9 eV with the other two above the surface (Fig. 4a). Isosurface plotting of the charge-density difference suggests strong charge depletion around the Ru atoms and the charge accumulation at the Ru$_6$-SRO interface (Fig. 4b). Plane-averaged charge-density difference along z-direction normal to the Ru$_6$/SRO interface suggests the significant changes in the charge densities around the Ru clusters (labeled as Ru1 site and Ru2 site in Fig. 4b), indicating the charge redistribution inside the Ru clusters (Fig. 4c). Hydrogen adsorption behaviors at different sites of the Ru$_6$/SRO surfaces are investigated. $\triangle G_{H^·}$ values are calculated to be −0.34 eV for the Ru2 site and −0.12 eV for the Ru1 sites (Fig. 4d, e and Table S6), which confirmed that the Ru atoms at the top of Ru$_6$ clusters are the active sites for HER. This value is comparable to the state-of-the-art Pt catalyst (−0.10 eV), and much smaller than that of the pristine SRO (Fig. 4e).

Further analysis indicates that the hydrogen adsorption behavior on the Ru2 sites is similar to the bulk phase of Ru, with a negative $\triangle G_{H^·}$ value of −0.45 eV[73]. Such negative adsorption energy suggests the

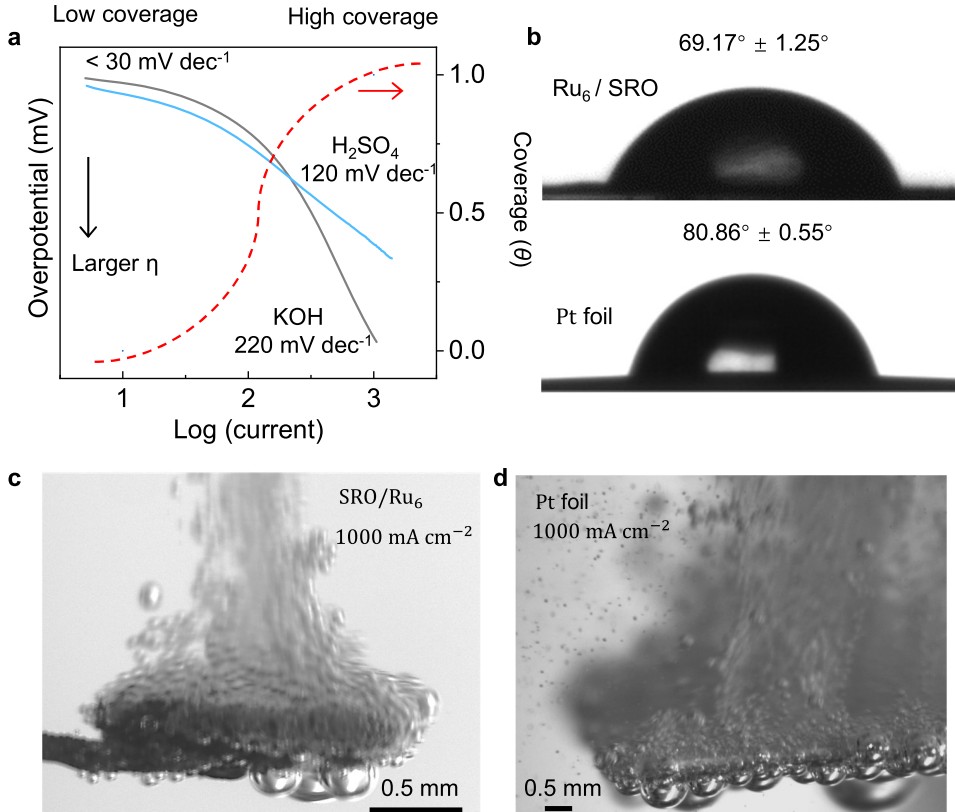

**Fig. 5 | Mechanism of the HER process at large current densities. a.** Tafel slope analysis of the $Ru_6$/SRO catalyst under high current densities. The red dotted line shows the coverage-dependent current densities. **b**. Water contact angles of $Ru_6$/SRO and Pt foil catalysts. The bubble evolution process on the surface of **c** Pt and **d** $Ru_6$/SRO catalysts at the current density of 1000 mA cm$^{-2}$. Bubble sizes are much smaller at the surface of $Ru_6$/SRO.

strong bonding between the catalyst and the hydrogen, which disfavors the subsequent desorption process and thus slows down the HER kinetics. On the other hand, the hydrogen adsorption is weakened at the Ru1 sites, making the HER process thermodynamically favorable. The reason can be found in the Bade charge transfer analysis, where more electrons are transferred from the Ru1 site ($0.29e^-$) to the pristine SRO than the Ru2 sites ($0.19e^-$) (Fig. 4d). This clearly indicates that the SRO support acts as a charge accumulation layer and the Ru sites are the catalytic active centers for HER. Benefiting from the fast electron transfer kinetics between the interface, a high reaction order is expected at the Ru sites, which leads to high performance at high current densities above 1000 mA cm$^{-2}$[74,75]. The relatively positive Ru atoms at the Ru1 site will weak the adsorption of the H reaction intermediate, promoting the Tafel step of hydrogen desorption, and finally resulting in high intrinsic HER efficiencies[76–78].

### The mechanism for the high efficiency at industrial-scale current densities

As we mentioned above, the working mechanisms of HER catalysts under low and high current densities are significantly different. The next important question is how is the reconstructed $Ru_6$/SRO catalyst related to the high performance at such a high current density above 1000 mA cm$^{-2}$. We have answered this question in the following three ways.

More information on the kinetics and mechanisms of the HER process can be obtained from the Tafel analysis. Thus, we re-examined the Tafel slopes at a large current range under acidic conditions (Fig. 5a). It is not strange to see the low Tafel slope of around 30 mV dec$^{-1}$ for our $Ru_6$/SRO catalyst and most reported Pt-like catalysts, which suggests that the recombination of adsorbed hydrogens is the rate-determination step. Here, one should be noted that there are no

electrons involved during this chemical reaction (Tafel step). However, the values of Tafel slope and reaction rates are strongly potential-dependent, and thus are dependent on the coverage of intermediate species ($\theta$) too[13]. Thus, for most catalysts, the Tafel slopes will increase rapidly with the increase of applied potentials, indicating that the catalytic activities are limited by electron transfer kinetics and adsorption of protons[14,79]. At a current density of ~1000 mA cm$^{-2}$, the Tafel slopes of $Ru_6$/SRO catalyst are determined to be 120 and 200 mV dec$^{-1}$ under acidic and alkaline electrolytes, respectively. It is interesting to see that these values are close to the state-of-the-art Pt catalysts, with Tafel slope as 125 mV dec$^{-1}$ at ~300 mA cm$^{-2}$ in 0.5 M $H_2SO_4$[19]. The value is close to the theoretical prediction by assuming coverage of $\theta = 1$, indicating the fast excellent hydrogen adsorption behavior at large current densities[13]. In addition, the low Tafel slopes guarantee relatively low applied voltage during industrial-scale applications. Another dominant factor that needs to be taken into consideration is the hydrogen bubble desorption behaviors. The fast release of produced bubbles at such a high reaction rate is vital for the mechanical stability of the catalysts, the continuous exposure of the active sites, the charge transfer kinetics, and the mass transfer efficiencies[20]. Thus, the hydrophilicity of the $Ru_6$/SRO catalyst surface is investigated and compared with the state-of-the-art Pt catalyst. The contact angle between the gas-liquid interface and the solid substrate is determined to be 69.2 and 80.9° for $Ru_6$/SRO and Pt foil, respectively (Fig. 5b). The improved wettability at the surface of $Ru_6$/SRO is favorable for the formation of uniform hydrogen bubbles with small size dimensions, which is critical for the fast bubble release and the depression of electrode polarization[20,21]. Indeed, digital image correlation (DIC) measurements under a current density of 1000 mA cm$^{-2}$ suggest that the bubble size at the surface of $Ru_6$/SRO (~ 50 μm) is much smaller than that on Pt foil (~100 μm) (Fig. 5c, d). Videos in slow motion further confirmed the growth and the release of hydrogen

bubbles is more rapid at the surface of $Ru_6$/SRO (Supplementary Movie 1) than at the Pt surface (Supplementary Movie 2).

The ultra-low Tafel slopes and the fast hydrogen bubble release kinetics are closely related to the unique structures of the reconstructed $Ru_6$/SRO catalyst. In contrast to the electrode designing strategies such as catalyst ink deposition or grown on the conductive substrate, the active Ru clusters are in situ formed at the surface of a highly conductive SRO single crystal. This will significantly decrease the charge transfer resistance at the interface, and, more importantly, boost the mechanical stability during the violent growth and release of hydrogen bubbles[80]. In addition, double-layer capacitance measurements suggest that the active surface areas are orders of magnitudes higher than their geometric area. This implies highly porous and 3D hierarchical structures of the Ru cluster layer, which is vital for the nucleation/release of hydrogen bubbles and high mass transfer efficiencies at large current densities.

In this work, we have developed a highly efficient and robust HER catalyst based on the in situ activation of SRO single crystals. Ferromagnetic Ru clusters are observed during the reduction process and serve as the catalytic active sites. In contrast to the bulk Ru phase, we found strong electron redistribution inside the Ru clusters as a result of the metal-support interaction. This leads to the optimized hydrogen adsorption behavior, which is thermodynamically favorable for HER. The highly conductive bulk phase as well as the interface could significantly decrease the charge transfer resistance and electrode polarization at high overpotentials. Aided by the high hierarchical surface-active layer, the reconstructed $Ru_6$/SRO catalyst exhibited remarkable HER activities and stabilities under both acidic and alkaline conditions. This work highlights that the tuning of the interface structures between the active phase and the substrate is critical for the designing of high-performance catalysts that are robust under industrial-scale current densities.

## Methods

### SRO single crystal growth

Single crystal of $Sr_2RuO_4$ samples were grown by the floating-zone method as described in the previous works[81]. Starting materials of $SrCO_3$ and $RuO_2$ with the molar ratio of 1: 0.6 were pressed into pallets and then sintered at 1150 °C for 24 h. The pallets were crashed into powder again and then filled into a balloon to shape the feed rod. The rod was then sintered again and then suspended in a floating-zone furnace. The feed speed is typically 28–30 mm/h, and the crystal-growth speed is 45 mm/h in a gas mixture of $O_2$ (15 or 10%) and Ar (85 or 90%) at 0.35 MPa. The feed and seed are rotated in opposite directions, typically at 30 rpm. Plate-like bulk single crystal with a geometry surface area of around 5 mm² was exfoliated from the parental crystal and used as the electrode for catalysis directly.

### Materials characterization

The SEM images were obtained by using a JEOL JSM 6700 F electron microscope with an accelerating voltage of 5 kV. A TITAN 80/300 electron microscope with an acceleration voltage of 200 kV was used to perform TEM, HRTEM, and corresponding EELS mapping. XPS analyses were performed by using a UHV surface analysis system equipped with a Scienta-200 hemispherical analyzer. The base pressure of a sample analysis chamber is $2 \times 10^{-10}$ mbar and corrected with the C 1s line at 284.6 eV. The standard deviation for the binding energy values was 0.1 eV. The bubbles evolution process was recorded by using digital image correlation (DIC). The contact angles were analyzed by dynamic contact angle measuring devices and a tensiometer (DCAT21). Measurement of the content of Sr and Ru in the reaction mixture by inductively coupled plasma emission spectrometer (ICP-OES) (SPECTRO ARCOS). Raman spectra were recorded using a customary confocal micro-Raman spectrometer with a HeNe-laser

(wavelength 632 nm) and a single-grating spectrograph with 1 cm⁻¹ resolution.

### Electrochemical measurements

Electrochemical measurements were performed on an Autolab PGSTAT302N electrochemistry workstation with an impedance module. A 150 mL commercial four-neck round bottom flask containing 100 mL 1 M KOH or 0.5 M $H_2SO_4$ aqueous solution (1 M) was used. Ag/AgCl (3 M KCl) and graphite rod are used as the reference electrode and counter electrode, respectively. 20% Pt/C (Sigma-Aldrich) commercial catalysts with a loading mass of 1 mg/cm² are used for comparison of catalytic activities. The SRO bulk single crystal was attached to the Cu wire with silver paint and served as the working electrode. All of the SRO crystals and part of the Cu wire were submerged in the electrolyte. For the measurement at 70 °C, the electrochemical cell was immersed in the water bath with built-in digital temperature control, with a Pt net as the counter electrode. Linear sweep voltammetry (LSV) curves and cyclic voltammetry (CV) curves are recorded with a speed of 1 and 50 mV/s, respectively. $i$R-compensation is performed by considering the solution resistance from the impedance spectra. Chronoamperometric tests were used to characterize the stability under room temperature and 334 K high temperature provided by the water bath. Electrochemical impedance spectroscopy (EIS) measurements were performed on an Autolab model 302 N potentiostat with a frequency range from 10 kHz to 0.1 Hz and a 10 mV AC dither. For Faradaic efficiency measurements, the hydrogen was collected and quantified by drainage using a measuring cylinder. An overpotential of about 280 mV is applied during the hydrogen bubble collection process. Hydrogen bubble formation and release kinetics were tracked by the Digital image correlation (DIC) technique (Correlation Solutions Inc.). The cameras were Rodenstock 105 mm f/5.6 Rodagon enlarging lens.

### Estimation turnover frequency calculations (TOF)

The electrochemical active surface area of the $Ru_6$/SRO catalyst is obtained by measuring the double-layer capacitance ($C_{dl}$) in the potential range with no faradic processes happening at various scan rates. The specific capacitance is determined by plotting the capacitive currents as a function of the scan rate.

The total number of hydrogen turn-overs was calculated from the current density according to equation (1):

$$\# H_2 = \left( j \frac{mA}{cm^2} \right) \left( \frac{1\,C\,s^{-1}}{1000\,mA} \right) \left( \frac{1\,mol\,e^-}{96485.3\,C} \right) \left( \frac{1\,mol\,H_2}{1\,mol\,e^-} \right) \left( \frac{6.022 \times 10^{23}\,H_2}{1\,mol\,H_2} \right)$$

$$= 3.12 \times 10^{15} \frac{H_2/s}{cm^2} \, per \, \frac{mA}{cm^2}$$

We assumed that all the Ru atoms at the top of the $Ru_6$ clusters are active towards HER. Thus the number of active sites per real surface area for Ru can be obtained by equation (2):

$$Ru \, \# \, active \, sites = \frac{1\,cm^{-2}}{\left( 2.25 \times 10^{-8}\,cm \right) \times \left( 2.34 \times 10^{-8}\,cm \right)} \times 2$$

Then the HER turnover frequency (TOF) as a function of current density is defined according to equation (3):

$$TOF = \frac{\left( 3.12 \times 10^{15} \frac{H_2/s}{cm^2} \, per \, \frac{mA}{cm^2} \right) \times |j|}{\# \, active \, sites \times A_{ECSA}}$$

## Data availability

All the data generated or analyzed during this study have been included in the manuscript and Supplementary Information. Source data

are provided as a Source Data file. Source data are provided with this paper.

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

## Acknowledgements

This work was financially supported by the National Natural Science Foundation of China (Grant No. 52271194 for G.L.), European Research Council (ERC Advanced Grant No. 742068 "TOPMAT" for C.F.), funding by the DFG through SFB 1143 (project ID. 247310070 for C.F.) and the Würzburg-Dresden Cluster of Excellence on Complexity and Topology in Quantum Matter ct.qmat (EXC2147, project ID. 39085490 for C.F.) and via DFG project HE 3543/35–1, and the foundation of the director of NIMTE, CAS. J.L. thanks the funding from the Zhejiang Provincial Natural Science Foundation of China (Grant No. LD21E010001), Ningbo Science and Technology Innovation 2025 Major Project (Grant No. 2020Z063). G.L. thanks Ulrich Burkhardt for taking the SEM measurements and elemental analysis. We also thanks Dmitry Knyazev for making the FIB sample and Hakan Deniz for the TEM measurements.

## Author contributions

G.L. and C.F. conceived and designed the experiments. The SRO crystals were provided by C.H., with the synthesis, physical properties characterization, and bulk crystallographic information determination performed principally by D.A.S., N.K., and K.E.A. Catalytic measurements, XPS analysis, SEM, DIC, and contact angle test are carried out by Y.Z. with the support of J.L. Theoretical investigations are performed by Q.Y. G.L. and Y.Z wrote the paper. K.E.A and Q.Y contribute equally to this work. All authors discussed the results and commented on the manuscript.

## Funding

## Competing interests

The authors declare no competing interests.
