## [Peer Review File · Nature Communications]

Observation of a robust and active catalyst for hydrogen evolution under high current densitiesREVIEWER COMMENTS

Reviewer #1 (Remarks to the Author):

This is a well written paper and the experiments are clear and well organized. There are a few questions to address and suggestions to improve the paper. Figure 1a, give a scale. Figure 1e, give error bars on all data points. Figures 2e, h, give error bars. Figure 3h, identify all peaks. Figures 4 a,b,d, give scales. Figure 4e, give error bars.

Reviewer #2 (Remarks to the Author):

This paper reports an interesting study on hydrogen evolution by water electrolysis at high current density, which is very important for large-scale hydrogen production. For alkaline electrolyzers, the current density is usually quite low due to high overpotentials for hydrogen evolution and oxygen evolution. So this research is of great interests to researcher working on electrolytic hydrogen production. Although the catalyst material has been reported by others already, this study provided insights in the mechanisms on the high current density and durable operation, by means of well designed characterizations/testing and DFT computations. After reading the paper, the reviewers is quite interested in this study and is happy to recommend acceptance of this paper for publication in Nature Communications after some revisions.

1. The authors confirmed that the high performance of the SRO catalyst is caused by the reconstructed Ru clusters on the SRO surface. This finding is not surprising, as the noble metal clusters on the support have been reported to have excellent performance as well. There are different ways of loading noble metal clusters on the support, for example, some researchers can load metals on the surface of perovskites materials by exsolution. Can the authors provide some discussions and comparisons on the different approaches for loading metals on the support for electrocatalysis? Is the approach reported in the present study better than others?
2. Some research works have been conducted on noble metal/support catalysts. It was found that the noble metal catalyst could be responsible for the high reaction order while the support could be responsible for accumulating positive charges. What's the role reaction order for HER in the study at a high current density? What's the role of Ru and SRO?
3. The testing was also conducted at 70 degree celsius. Then what's the effect of operating temperature on the hydrogen evolution kinetics? What's the effect of temperature on the bubbling behaviours? Would the temperature affect the Ru cluster formation process? And how?
4. The formation of Ru₆/SRO will affect the material properties of SRO as well. Is the activity and/or conductivity of SRO affected by Ru₆?
5. When comparing the results with the literature data, the authors may also provide the electrochemically active surface area of the catalyst.
6. What's the optimal morphology of Ru₆/SRO to enable fast bubble desorption? Maybe the authors can consider conduct multiphysics modeling of the bubbling behavior in the future research.
7. The authors only studied hydrogen evolution reaction, so it is not a complete electrolyzer for water-splitting. It could be more useful if a complete electrolyzer is constructed for hydrogen and oxygen production by water splitting.

Reviewer #3 (Remarks to the Author):

Yudi Zhang and coworkers report a highly active electrocatalyst for the hydrogen evolution reaction capable of reaching high current densities up to 1000 mA/cm² both in alkaline and acidic environment for hundreds of hours. The catalyst is based on Sr₂RuO₄ single crystal, which after activation, a Ru migration towards the surface is observed, forming Ru islands. These Ru islands are considered the active sites. The high conductivity, wettability, and near perfect hydrogen-interaction energies are considered to be the source of such high activity.

The reported electrocatalyst contains Ru (14.2 At%), and so such high current densities are expected. In addition the Ru migration towards the surface suggest that there is plenty of Ru available for the HER. Therefore the HER activity should be normalized again the available Ru to have a better comparison with Ru-based electrocatalysts. In addition to this comment, there are other issues that should be addressed:

-I suggest to adapt the title to indicate that only the hydrogen evolution reaction is being studied, since water electrolysis also involves the OER not considered in this study.

-Pt is being used as counter electrode during the stability test. This is completely unacceptable, it is well known that Pt will slowly re-deposit onto the cathode, specially under high current densities and the long-time operation. Pt deposition has been observed in just few hours at -50 mA/cm^2 , therefore I would expect that deposition of Pt would be more significant in this case. The authors needs to verify that no Pt has been deposited by proper accurately techniques.

- Ru is known to be highly active towards HER, so in this case it is not surprising the observed high current density. Therefore, it is important to know the catalyst loading in term of Ru content. The catalyst is stated as single crystal, but even if the reaction is only carried at the surface, there is Ru migration from the entire bulk towards the surface. So the current density should be stated in term of the entire Ru loading. Only in this case the catalyst can be compared with other Ru-based catalysts.

-Due the extreme test conditions, it is needed to indicate the characteristic on the electrochemical cell. How is the temperature of the cell controlled? Is it a H-cell? Volume of the cell? Is the electrolyte being recirculated? Similarly, detailed description (added in supporting information if needed) on how the electrocatalysts is placed into the cell, Is the wire in contact with the electrolyte? Is the electrocatalysts partially submerged? What it the HER contribution of the wire and silver paint?

-Using DFT to evaluate the HER activity in cluster-based systems is challenging. The correct configuration is not known, as well as the H-coverage. Therefore several cluster configurations should be tested even if some have slightly larger formation energies than the "most energetically favorable cluster". It would not be correct to assume that all clusters will be in the most stable configuration when the surface of the SRO catalyst is being reconstructed. However in this case, only a single Ru₆ cluster was selected, this easily creates a bias since the most convenient systems tends to be reported.

-The DFT study uses a Ru₆ cluster as catalysts, but why is the Ru₆ used? Is there any experimental evidence that this is the case? From SEM there seems to be much larger particle formation. How much Ru is then available at the surface?

-The authors indicate that
"The relatively positive Ru atoms at the Ru1 site will weak the adsorption of the H reaction intermediate, promoting the Tafel step of hydrogen desorption, and finally resulting in the high intrinsic HER efficiencies."

Could the authors provide the reasoning behind this? This is contrary to what is being reported for molecular catalysts, N-doped nanocarbons, and sulfur-rich MoS₂, where a slightly negative charge on the active site (due to doping) is beneficial for the HER since the active site will easily allow the electron transfer to the proton.

-The authors uses a fairly complex equivalent circuit diagram during the EIS analysis, but there is no information regarding the meaning of it. The authors should provide further details. What is the origin of the two semicircles?

Some other minor comments:

-There seems to be a mistake in the wire used to connect the catalysts. The experimental section indicates Cu, but in the "HER catalytic performance" section indicates Ti.

-The specific surface area of the SRO is mentioned in the main text, and in Supp information (Figure S5) but no value is given. This information should be specified.

- How much of the electrolyte resistance was corrected?

-Line 181 in the submitted manuscript, Fig. 2E is mentioned, but it should be 2F, when mentioning the 9000 mA/cm².

-Indicate the methodology used to measure the bubble size.

-For the DFT computations. When evaluating the Gibbs free energy, the zero point energy and entropy changes are required. But no information is provided about these quantities. Clearly indicate how these were evaluated.

-Given the value of the adsorption free energy, a near complete Hydrogen monolayer is expected, but only a single H atom is used during the evaluation. This can significantly affect the hydrogen-Ru interaction.

Response to reviewers

Reviewer #1 (Remarks to the Author):

This is a well written paper and the experiments are clear and well organized. There are a few questions to address and suggestions to improve the paper. Figure 1a, give a scale. Figure 1e, give error bars on all data points. Figures 2e, h, give error bars. Figure 3h, identify all peaks. Figures 4 a,b,d, give scales. Figure 4e, give error bars.

Response: We greatly appreciate your positive evaluation and kind suggestions. To get the error bars, we analyzed data by carrying out more parallel measurements or checking the error limit of the relevant equipment.

In Figure 1a, we include the scale by considering the lattice parameters of Sr₂RuO₄.

*In Figure 1e, we analyzed the resistivity data of **three independent measurements** for SRO, and make three independent measurements on the re-constructed SRO after catalysis. The conductivity data for other reference materials such as Ir, Ru, and Pt, are taken from previous reports. We included the reference for each compound.*

*In Figure 2e, we took **a new crystal and then activated it with the same procedure**. LSV curves were recorded two times (**Figure R1**). The overpotentials are obtained and then compared with the initial measurements at the current density of 1000 mA cm⁻². **Error bar is given based on these three LSV measurements**. The data for other materials are taken from published data and thus we only include the used references.*

Figure R1. LSV curves of a new activated SRO crystal in 1M KOH electrolyte.

In *Figure 2h*, we include the error bar by checking *the measuring accuracy* of the graduated cylinder at 20 °C.

In *Figure 4 a, b, and d*, we give the scale by considering the lattice parameters of the used unit cell.

In Figure 4e, the Gibbs free energy is based on the DFT calculations on a fixed crystal structure and hydrogen adsorption geometry. We inquired about colleagues who performed the calculations and then checked other published works. It is found that the *values will not be changed except when one changes the used slabs for calculations*.

We once again thank you for your time to review our work and hope you will be satisfied with our response.

Reviewer #2 (Remarks to the Author):

This paper reports an interesting study on hydrogen evolution by water electrolysis at high current density, which is very important for large-scale hydrogen production. For alkaline electrolyzers, the current density is usually quite low due to high overpotentials for hydrogen evolution and oxygen evolution. So this research is of great interests to researcher working on electrolytic hydrogen production. Although the catalyst material has been reported by others already, this study provided insights in the mechanisms on the high current density and durable operation, by means of well-designed characterizations/testing and DFT computations. After reading the paper, the reviewer is quite interested in this study and is happy to recommend acceptance of this paper for publication in Nature Communications after some revisions.

Response: At first, we deeply appreciate you for the encouraging comments and suggestions, which are constructive for us to improve our manuscript. We revised the manuscript extensively according to your comments and hope that you will be satisfied with our response.

1. The authors confirmed that the high performance of the SRO catalyst is caused by the reconstructed Ru clusters on the SRO surface. This finding is not surprising, as the noble metal clusters on the support have been reported to have excellent performance as well. There are different ways of loading noble metal clusters on the support, for example, some researchers can load metals on the surface of perovskites materials by exsolution. Can the authors provide some discussions and comparisons on the different approaches for loading metals on the support for electrocatalysis? Is the approach reported in the present study better than others?

Response: Thanks for pointing out this important issue in catalyst design. As the reviewer suggested, there is extensive literature on noble metals loading on the supports. Most of them have been confirmed to be effective in boosting catalytic efficiencies. Here we briefly introduce and compare three commonly used strategies.

One of the most used methods is the drop-casting technique (A. Kumar, et al. *Electrochem. Commun.* 2020, 121, 106867). In this method, catalyst inks including noble metals nanostructures should be made, and then dispersed into various support

such as carbon paper, Ni foams, or other conductive substrates. ***This is a versatile and cost-effective method*** for loading active catalysts but with obvious ***disadvantages such as the coffee-ring effect and uncontrolled aggregation*** of the noble metal nanostructures (Li et al. *Chem. Sci.*, 2018, 9, 7596). In addition, inconsistent results are always obtained depending on the details and skill of the researchers even though all the experiment details are kept the same. For example, when used as a hydrogen evolution reaction catalyst in 0.5 M H₂SO₄, the overpotentials for the commercial 20 % Pt/C catalysts could be as low as 20 mV (Li et al. *Adv. Funct. Mater.* 2018, 1803722), or up to 70 mV (Zheng et al. *Nat Commun.* 2014, 5, 3783)

To address the aggregation and inhomogeneous of the loading metals, ***gas-phase deposition methods*** have been developed such as atomic layer deposition, and chemical vapor deposition (Pan, et al. *Nat Commun.*, 2022, 13, 2294; Zhang et al. *Natl. Sci. Rev.*, 2018, 5, 1). Benefiting from the ***strong affinity between the loading metals and supports***, charge transfer and quantum hybridization at the interface can be effectively tailored to increase catalytic efficiencies. However, ***the expensive synthesis facilities, noble metal precursors, and complicated operation procedures*** restricted the practical applications. Other detrimental including fast deactivation and agglomeration because of the harsh and critical experimental conditions such as high temperature and high applied potentials during catalysis processes (Liu, et al. *Chem. Rev.* 2018, 118, 4981).

Recently, ***exsolution has been proposed as a promising method to load noble catalysts***. As the reviewer pointed out, this method gains great success in producing active metal catalysts from the perovskite oxide lattices (Opitz, et al. *Nat Commun.*, 2020, 11, 4801; Li et al. *Chem. Soc. Rev.*, 2017, 46, 6345; Tang et al. *Nanoscale*, 2019, 11, 16935). This can be viewed as an indirect way for catalysts loading because the active metal clusters/nanostructures are segregated from the bulk oxide to its surface under a reducing atmosphere. In this method, the exsolved metal nanostructures are socketed into the oxide bulk phases, ***leading to enhanced cohesion between the produced metal catalysts and the supports***. This guarantee the ***well-dispersion and good stability*** of the modified catalysts, favoring the electrochemical performance because of the strong interaction and the accelerated electron transfer kinetics between

the interface (Kwon et al. *J. Phys. Energy*, 2020, 2 032001; Yu et al. *Nano Lett.*, 2020, 20, 3538). In addition, compositionally and size-diverse nanoparticles can be obtained during the exsolution process, including metal, metal oxides, or their mixtures from clusters to particles up to sever tens nanometers (Neagu, et al. *Nat. Chem.*, 2013, 5, 916).

In this work, we use an *in-situ* electrochemical reduction method to produce active metal catalysts. In essence, this can be also viewed as an exsolution process as the reviewer commented. In addition to the advantages of enhanced cohesion and favored electron transfer kinetics, we also found *excellent chemical stabilities* of the in-situ formation Ru clusters. This can be confirmed by the fact that no obvious Ru aggregation is observed under harsh conditions and after a high current density test. Besides, this is a *fast and efficient method* in comparison with previously reported works. The current density is increased four times after a cyclic voltammetry method of 30 cycles (less than two minutes, Figure 3a).

In the revised manuscript, we discussed these different strategies of metal catalyst loading in the introduction section.

2. Some research works have been conducted on noble metal/support catalysts. It was found that the noble metal catalyst could be responsible for the high reaction order while the support could be responsible for accumulating positive charges. What's the role reaction order for HER in the study at a high current density? What's the role of Ru and SRO?

Response: We thank the learned reviewer to mention this important parameter to understand catalysis kinetics. If we understand correctly, the reaction order here means the change of current density as a function of reactant activity at a given constant voltage, which is obtained from rate law analyses. (Formal, et al. *J. Am. Chem. Soc.* 2015, 137, 6629; Dai et al. *Adv. Funct. Mater.* 2022, 32, 2111989).

As the reviewer suggested, the determination of the reaction order guides identifying the inherent reaction mechanisms at high current densities/reaction rates. This has gained great success in understanding the highly efficient metal/support catalysts for multi-electron transfer reactions (Li, et al. *Nat Commun.*, 2021, 12, 255; Dai et al. *Adv.*

Funct. Mater. 2022, 32, 2111989; Schuler, et al. *Energy Environ. Sci.*, 2020,13, 2153-2166). A good catalyst requires, essentially, a pseudocapacitive feature close to the OER potential for the high charge accumulation/hole densities at the active sites. We tried to analyze the reaction order with the above-proposed strategy but found difficulty in this work. The main reason is that when we use pulse voltammetry to quantify the accumulated positive charge, the aimed and open circuit potential overlap with the hydrogen underpotential deposition of the reconstructed SRO/Ru catalyst. Most effective metal catalysts show strong HUD peaks near the HER region. This may be the reason why we did not see many reports on the analysis reaction orders of HER catalysis with this method.

Figure R2. a. Bode plots of SRO catalyst before and after HER testing, 20 % Pt/C, and Ru SAC (Wu et al *Inorg. Chem.* 2022, 61, 11011). b. illustration of the role of Ru clusters and SRO support.

However, *inspired by the peak frequency analysis of the EIS-Bold plot (Dai et al. Adv. Funct. Mater.* 2022, 32, 2111989), we can expect high reaction orders of the reconstructed SRO/Ru catalyst at high current density. **Figure R2a** displayed the EIS-Bold plots of SRO/Ru catalyst before and after activation, 20 Pt/C catalysts, and high-performance Ru single-atom catalysts (Wu et al *Inorg. Chem.* 2022, 61, 11011). The Bode plots deliver a tendency of SRO/Ru activated (25 Hz) > SRO/Ru fresh (20 Hz) > Ru SAC (12 Hz) > 20 Pt/C (1.6), *suggesting the fast electron transfer kinetics and*

high reaction order of the reconstructed catalyst. We also analyzed the Bode charge transfer between the Ru clusters and the SRO support. The electron transfer from the Ru1 site and Ru2 site to the pristine SRO support are $0.29e^-$ and $0.19e^-$ respectively (Fig. 4d), suggesting the *positive charge accumulation at the support.*

So far, we can conclude that the SRO support act as a charge accumulation layer and the Ru sites are the catalytic active centers for HER (**Figure R2b**). When Ru clusters become slightly positive, the binding energy of hydrogen can be weakened in comparison with the Ru metal, thus favoring the hydrogen desorption process.

Although we cannot get the specific reaction orders, *we still discussed their vital roles in the revised manuscript (Bode analysis and Bode charge transfer section).* We hope we can find a suitable system to apply this strategy in the analysis of HER catalysts in the future.

3. *The testing was also conducted at 70 degree celsius. Then what's the effect of operating temperature on the hydrogen evolution kinetics? What's the effect of temperature on the bubbling behaviors? Would the temperature affect the Ru cluster formation process? And how?*

Response: We thank the reviewer for this good suggestion. To understand the catalytic behaviors of the developed catalyst at higher temperatures, we repeated the catalytic measurements and analyzed the structure changes and bubble release process.

LSV curve was recorded by using Ag/AgCl (3.5 M KCl) as the reference electrode and a graphite rod as the counter electrode in the 1 M KOH electrolyte. To reach a current density of 1000 mA cm^{-2} , the overpotential was decreased from 354 mV at RT to only ~241 mV at 70 °C (**Figure R3a**). Accordingly, the overpotential difference before and after iR correction was decreased to only ~ 10 mV, suggesting *accelerated electron transfer kinetics at higher temperatures.* This can be further confirmed by the decrease in both charge transfer resistance and ohmic resistance from the impedance measurement at 70 °C (**Figure R3b**). This is not strange as the activation energy is significantly decreased with the increasing of temperature according to the Arrhenius equation.

Next, we turned to the analysis of Tafel slopes. Tafel slopes were determined to be 27 and 232 mA dec^{-1} (**Figure R3c**) around the low and high current density regimes, respectively, which are close to that of room temperature, *implying that hydrogen evolution kinetics is not sensitive to temperature*. This means that the finite temperature increase ($< 50\text{ }^{\circ}\text{C}$ in this work) will not change the electronic structures and adsorption behaviors of our sample. We also monitored the hydrogen bubble release process at $50\text{ }^{\circ}\text{C}$ with the digital image correlation (DIC) technique. *There are no significant changes in both bubble size and distribution in comparison to the catalysis at RT (Figure R3d).*

Figure R3. (a) LSV curves of the SRO/Ru catalyst with (w) and without (w/o) considering the ohmic drop at 70°C . (b) Comparison of the Nyquist plots of SRO/Ru catalyst before and after HER testing at room temperature and 70°C . (c) Tafel slope analysis of the SRO/Ru at 70°C . (d) Hydrogen bubble release process at 70°C .

We analyze the structural changes after a high-temperature catalysis reaction. SEM images show that the crystal surface experienced obvious reconstruction with increased roughness (**Figure R4 a-b**), which is *almost the same as the catalysts after the reaction at RT*. EDS pattern (**Figure R4 c**) recorded at the surface suggests more Ru elements

at the surface, with the Sr : Ru ratio decreasing from the ideal 2:1 to about 0.8:1. Elemental mapping (Figure R4 d-e) at the cross-section of the plate-like crystal suggests that the edges are Ru-rich, with significant Sr losing. Finally, XPS measurements (Figure R4 f) were taken at the crystal surface after the reaction. Survey spectra suggest the existence of Sr, O, Ru, and Ag elements, where the Ag element is from the silver paint for making the contact. We fitted the Ru spectrum with similar parameters as we understand the RT catalytic behaviors. It can be seen that the strong peak with a binding energy of 280.1 eV can be attributed to Ru (0) metal. *This suggests the formation of Ru metal layer at the reconstructed surface.* All the above measurements indicate that *temperature has little effect* on the surface reconstruction process during catalysis reactions.

We include the analysis of the structural changes after high-temperature measurement and discussed them briefly in the revised manuscript.

Figure R4. a and b: SEM images of the SRO crystal surface after long-term testing at 70°C. c. EDS spectrum recorded at the same crystal after catalysis. Elemental mapping of the cross section d. Ru element, and e. Sr element. f. Ru 3d spectrum of the SRO crystal after catalysis.

4. *The formation of Ru6/SRO will affect the material properties of SRO as well. Is the activity and/or conductivity of SRO affected by Ru6?*

Response: We thank the reviewer for this nice suggestion. Indeed, the conductivity of the catalysts during and after catalytic measurements are also important parameters in understanding catalytic properties. We thus measured the reconstructed SRO/Ru catalysts after long-time measurement at the high current density of 1000 mA cm^{-2} . *The conductivity was measured three times with the four-point probes technique at room temperature*, which is determined to be 2.7×10^5 , 2.6×10^5 , and $3.0 \times 10^5 \text{ S/m}$, respectively (**Fig. R5**). Although they are a slight decrease in conductivity in comparison with the perfect SRO crystal without surface modification, it still set SRO/Ru as one of the most conductive catalysts.

We included the conductivities of the reconstructed catalysts and discussed them briefly in the revised manuscript.

Figure R5. Comparison of the fresh and re-constructed SRO catalysts with reported state-of-the-art catalysts.

5. *When comparing the results with the literature data, the authors may also provide the electrochemically active surface area of the catalyst.*

Response: We agree with the reviewer that the comparison of electrochemically active surface areas is necessary for understanding the intrinsic activities of catalysts.

For the reconstructed SRO/Ru catalyst, the electrochemical active surface area (ECSA) was determined by measuring the double-layer capacitance (C_{dl}) in the non-faradaic region. The C_{dl} and ECSA are determined to be 10 mF cm^{-2} and 250 cm^2 for the reconstructed catalysts (Figure S5). We compared the value with the ECSAs of state-of-the-art catalysts that we referred to in this work (**Fig. R6**), including MoC–Mo₂C-790 (2778 cm^2 , *Nat. Commun.*, 2021, 16, 6776), Ru-CoO_x/NF (2117 cm^2 , *Small*,

2021, 17, 2102777), and Ru-1.0 single atoms (1958 cm², *Nat. Commun.*, 2022, 13, 3958). Although the ECSA of our SRO/Ru catalyst is much smaller, it still exhibits high apparent catalytic activities, which further confirms the high intrinsic activities after reconstruction.

We include the comparison of ECSAs and briefly discussed it in the revised manuscript.

Figure R6. Comparison of the ECSAs between the developed SRO/Ru catalysts with recently reported state-of-the-art catalysts.

6. *What's the optimal morphology of Ru6/SRO to enable fast bubble desorption? Maybe the authors can consider conduct multi-physics modeling of the bubbling behavior in the future research.*

Response: We thank the reviewer to point out this essential problem in understanding HER mechanism.

Indeed, researchers in the early stage do not consider bubble desorption problems so much, but they received increasing attention now. A common understanding is that *the morphology and surface properties of the catalysts should favor the formation of small-size hydrogen bubbles and the subsequent fast detachment kinetics*. For this purpose, the surface of the catalysts should have fewer crevices or defects such as impurities, to make sure the *homogeneous nucleation of hydrogen bubbles* (Alchley et al. *J. Acoust. Soc.*, 1989, 86, 1065). Then, fast hydrogen bubble detachment is also expected to avoid further growth. It is reported that a *hydrophilic surface* is essential for the good wetting of the catalysts, which is beneficial for the fast liquid electrolyte

and gas transfer (Luo et al. *Nat Commun.*, 2019, 10, 269; Angulo et al. *Joule*, 2020, 4, 555). Finally, *good hydrophilicity between the catalyst surface and gas bubbles* is also necessary for the small bubble adhesive forces and smaller residency time (Hao et al. *ACS Appl. Energy Mater.*, 2019, 2, 5734).

Luckily, the reconstructed SRO/Ru catalysts have most of the above characteristics. A dense Ru clusters layer, rather than a polycrystalline nanoparticle formed at the SRO surface, leading to a *homogeneous distribution of active sites*. This is beneficial for the homogeneous nucleation of hydrogen bubbles and their unpinning at the catalyst surface. Most importantly, the in-situ formed *Ru clusters are positive charges* because of the charge accumulation at the SRO support. This may explain the observed *hydrophilic property of the catalyst surface* (Qi et al. *Chemosphere*. 2018, 202: 105).

We thank the learned reviewer for the suggestion of using multi-physics modeling, which is more powerful in the understanding of bubbling behavior. However, due to the fast dynamics and difficulties to visualize the small scale, it is a challenge for us and our collaborators to apply this method. We will consider this strategy in the future according to the reviewer's suggestion.

7. The authors only studied hydrogen evolution reaction, so it is not a complete electrolyzer for water-splitting. It could be more useful if a complete electrolyzer is constructed for hydrogen and oxygen production by water splitting.

Response: We thank the reviewer for this detailed suggestion. Indeed, we only focus on the HER properties of the SRO/Ru catalyst in this work. We revised the title to “*Observation of a robust and highly active catalyst for hydrogen evolution under the current density of 1000 mA cm⁻²*”.

Figure R7. (a) LSV curves of overall water electrolysis with the SRO as the cathode and commercial Ir/C as the anode in 1M KOH and the current density reaches 1000 mA cm⁻². (b) Long-term test of water electrolysis with SRO as the cathodes, and commercial Ir/C as the anode, indicating high electrochemical stability.

Per the reviewer's suggestion, *a complete electrolyzer was assessed* by using the developed SRO/Ru catalyst as the cathode and the commercial 10% Ir/C powder (Aladdin) as the anode. The Ir/C (loading mass 1 mg/cm²) was anchored to a Ni foam using Nafion. The water electrolysis was studied in 1 M KOH. The SRO /Ru|| Ir/C catalyst requires a potential of 1.87 V to reach the current density of 1000 mA cm⁻² when considering iR drop (**Figure R7a**). This electrolyzer also exhibited excellent water-electrolysis stability with negligible decay for over 24 h when operating at large current densities of 1000 mA cm⁻² (**Figure R7b**).

We revised our manuscript accordingly and discussed the overall water-splitting properties briefly in the revised manuscript. We once again thank the nice suggestions from the reviewer and wish you will be satisfied with our response.

Reviewer #3 (Remarks to the Author):

Yudi Zhang and coworkers report a highly active electrocatalyst for the hydrogen evolution reaction capable of reaching high current densities up to 1000 mA/cm² both in alkaline and acidic environment for hundreds of hours. The catalyst is based on Sr₂RuO₄ single crystal, which after activation, a Ru migration towards the surface is observed, forming Ru islands. These Ru islands are considered the active sites. The high conductivity, wettability, and near perfect hydrogen-interaction energies are considered to be the source of such high activity.

Response: We deeply appreciate you for taking the time to review our manuscript and providing these valuable suggestions. We have considered your all suggestions carefully and made point-by-point responses as shown below. We hope that you will be satisfied with our response.

1. The reported electrocatalyst contains Ru (14.2 At%), and so such high current densities are expected. In addition the Ru migration towards the surface suggest that there is plenty of Ru available for the HER. Therefore the HER activity should be normalized again the available Ru to have a better comparison with Ru-based electrocatalysts. In addition to this comment, there are other issues that should be addressed:

Response: We thank the reviewer for this suggestion. Indeed, mass activity is one of the fairest parameters to assess the intrinsic catalytic activities and their costs. We thus calculated the mass activities of the reconstructed catalyst and compared it with the recently reported state-of-the-art catalysts. The calculated details are described as followed:

Although the thickness of the used bulk single crystal is up to 140 μm (**Figure R 8a**), we found that the activation depth is only about 30 nm after the stability test (**Figure R 8b**). We assumed that all the material in this layer is pure Ru metal. By *considering the density of Ru* (<https://www.rsc.org/periodic-table/element/44/ruthenium>) and the current density in 1 MOH electrolyte (**Figure R 8c**), *the mass activity of SRO/Ru cluster catalyst is 16.6 A mg⁻¹ at an overpotential of 50 mV*. This value sets the SRO/Ru catalyst as one of the most competitive noble metal-based catalysts (Figure 2f, **Figure R12**).

Here, we should admit that *the contribution of SRO support is neglected*. With the bulk single crystals, we determined the real active sites come from the surface Ru clusters, while the SRO support acts as a charge-accumulating layer. We proposed that *the use of Ru elements can be significantly decreased by making thin films or nanostructures*, which is the emphasis of our future work.

We include the discussion on mass activity and its detailed calculation method in the revised manuscript. A brief perspective on how to decrease the Ru loading is given.

Figure R8. (a) SEM image of the cross-section of the SRO bulk single crystal. (b). TEM image of the activated SRO bulk single crystal. The depth of the in-situ Ru layer is estimated to be 30 nm. (c) Parameters to calculate the mass activity of the reconstructed Ru cluster.

2. I suggest to adapt the title to indicate that only the hydrogen evolution reaction is being studied, since water electrolysis also involves the OER not considered in this study.

Response: We thank the reviewer for this good suggestion. Indeed, we mainly investigated the HER behaviors of the bulk single crystal. We revised the title to “*Observation of a robust and highly active catalyst for hydrogen evolution under the current density of $1000\ mA\ cm^{-2}$* ”.

To check its performance in full water splitting, a complete electrolyzer was assessed by using the developed SRO/Ru catalyst as the cathode and the commercial 10% Ir/C

powder (Aladdin) as the anode. The Ir/C (loading mass 1 mg/cm²) was anchored to a Ni foam using Nafion. The water electrolysis was studied in 1 M KOH. The SRO /Ru|| Ir/C catalyst requires a potential of 1.87 V to reach the current density of 1000 mA cm⁻² when considering iR drop (**Figure R9a**). This electrolyzer also exhibited excellent water-electrolysis stability with negligible decay for over 24 h when operating at large current densities of 1000 mA cm⁻² (**Figure R9b**).

Figure R9. (a) LSV curves of overall water electrolysis with the SRO as the cathode and commercial Ir/C as the anode in 1M KOH and the current density reaches 1000 mA cm⁻² (iR corrected). (b) Long-term test of water electrolysis with SRO as the cathodes, and commercial Ir/C as the anode, indicating high electrochemical stability.

However, the main aim of this work is to investigate the HER behaviors, we changed the title according to your suggestions and then briefly discussed the full water-splitting performance in the revised manuscript.

3. Pt is being used as counter electrode during the stability test. This is completely unacceptable, it is well known that Pt will slowly re-deposit onto the cathode, specially under high current densities and the long-time operation. Pt deposition has been observed in just few hours at -50 mA/cm², therefore I would expect that deposition of Pt would be more significant in this case. The authors need to verify that no Pt has been deposited by proper accurately techniques.

Response: We thank the learned reviewer for this valuable suggestion.

Indeed, the observation of Pt deposition at the cathode surface has been reported by

different research groups (Bird et al. *ACS Appl. Mater. Interfaces*, 2020, 12, 18, 20500). Thus, the electrochemical stability test with Pt as the counter electrode should be interpreted carefully.

In this work, *all the electrochemical measurements at room temperature are carried out by using a graphite rod as the counter electrode, including the stability tests at 1000 mA cm^{-2} in acid and 1000 mA cm^{-2} in 1M KOH electrolyte* (Figure 2g in the main manuscript, **Figure R10**).

Figure R10. A photo was taken during the stability test. A graphite rod was used during the measurements (Taken on 3rd Dec. 2021, more details- can be provided in the request).

A Pt counter electrode was used during the stability test at 70 °C. The reason is that we found a quick graphite rod broken under these harsh conditions, with the observation of black powders at the cell bottom. To check whether the use of a Pt counter electrode influences the stability test, we make further analysis according to the suggestions from the reviewer.

Figure R11. (a) XPS survey spectrum of SRO/Ru catalyst after stability test at 70 °C. (b) XPS spectrum of the Pt 4f component.

XPS is the most surface-sensitive technique to check the surface elements' distribution. *We did not see a significant contribution from the Pt element* either from the survey spectrum or the scanning in the Pt 4f binding energy regime (**Figure R11**). This means that no Pt deposition happens at the cathode or is lower than the detection limit. This can be further confirmed by the relatively stable current density during the measurements, without a significant increase in catalytic activities (Figure 2g). We estimated that *the suppressed Pt dissolution may be a result of the small surface areas of the tested bulk single crystal*, which is only about 0.05 cm². Previous works found that the relatively small working electrode will lead to a small range of potential on the Pt counter electrode, and thus decrease the dissolution of Pt (Chen et al. *ACS Energy Lett.* 2017, 2, 5, 1070).

However, we strongly agree with the reviewer that any catalytic assessments using Pt as the counter electrode should be avoided or treated carefully, especially when increased activities are reported during the measurements.

4. *Ru is known to be highly active towards HER, so in this case it is not surprising the observed high current density. Therefore, it is important to know the catalyst loading in term of Ru content. The catalyst is stated as single crystal, but even if the reaction is only carried at the surface, there is Ru migration from the entire bulk towards the surface. So the current density should be stated in term of the entire Ru loading. Only in this case the catalyst can be compared with other Ru-based catalysts.*

Response: We agree with the reviewer that the comparison of mass activities is meaningful and necessary to highlight their potential commercial applications.

We assume that the Ru loading mass equals the in-situ formed Ru clusters at the crystal surface. The mass activities of the reconstructed SRO/Ru catalyst can be calculated according to the HER current and mass loading at a given overpotential (calculation *details can be seen in response to question 1*). The mass activity of SRO/Ru is determined to be 16.6 A mg⁻¹ at an overpotential of 50 mV, making it one of the best noble-metal-based HER catalysts although still not as good as some Ru-based nanostructures (**Figure R12**). We should admit that *bulk single crystals are the ideal platform to study the catalysis behaviors and mechanisms, but nanostructures are preferred to decrease the cost.*

Figure R12. Comparison of mass activities between the reconstructed SRO/Ru and state-of-the-art noble metal based HER catalysts.

We include the comparison of mass activities in the revised manuscript (**Figure 2f**) and discussed it accordingly.

5. Due the extreme test conditions, it is needed to indicate the characteristic on the electrochemical cell. How is the temperature of the cell controlled? Is it a H-cell? Volume of the cell? Is the electrolyte being recirculated? Similarly, detailed description (added in supporting information if needed) on how the electrocatalysts is placed into the cell, Is the wire in contact with the electrolyte? Is the electrocatalysts partially submerged? What it the HER contribution of the wire and silver paint?

Response: Thanks to the reviewer for this good suggestion. We are sorry for missing the detailed experimental conditions.

For the electrochemical measurements, a 150 mL commercial four-neck round bottom flask containing 100 mL KOH aqueous solution (1 M) was used. We did not perform electrolyte recirculation due to the limitation of equipment. However, we refill and replace the electrolyte several times, to make sure stable measurement conditions. The SRO bulk single crystal was attached to the Cu wire with silver paint and served as the working electrode. All of the SRO crystals and part of the Cu wire were submerged in the electrolyte. For the measurement at 70 °C, the electrochemical cell was immersed in the water bath with built-in digital temperature control. We did not see obvious water evaporation during the measurement as we covered it with plastic

wrap.

As part of the Cu wire and the silver paint were submerged in the solution, we measured their contributions to the HER current (**Figure R13**). At the overpotentials to reach the current density of 1000 mA cm^{-2} , the absolute current from the Cu wire and silver paint are determined to be 0.04 mA, 0.09 mA, and 0.5 mA, which takes 0.08 %, 0.18 %, and 1 % of the total current in 0.5 M H_2SO_4 (RT), 1 M KOH (RT) and 70°C . Thus, we can conclude that the high HER activities are derived from the reconstructed SRO catalyst.

We include all the above-discussed details in the experimental section in the supporting information.

Figure R13. LSV curves of Cu and silver paint at different measurement conditions (room temperature in 1 M KOH and 0.5 M H_2SO_4 , and 70°C in 1 M KOH).

6. Using DFT to evaluate the HER activity in cluster-based systems is challenging. The correct configuration is not known, as well as the H-coverage. Therefore several cluster configurations should be tested even if some have slightly larger formation energies than the “most energetically favorable cluster”. It would not be correct to assume that all clusters will be in the most stable configuration when the surface of the SRO catalyst is being reconstructed. However in this case, only a single Ru₆ cluster was selected, this easily creates a bias since the most convenient systems tends to be reported.

Response: We thank the reviewer for pointing out this important problem in theoretical understanding. Although we think that Ru₆ is the most possible cluster configuration (see the response to question 7), we agree with the reviewer that one cannot exclude the

existence of other types of clusters. To better understand the influence of Ru geometry and HER activities, *we investigated two other Ru configurations, including an ultra-small cluster of Ru 1 (Ru single atom) and a relatively large Ru 10 cluster (Figure R 14a and e)*. The adsorption energy of Ru1 and Ru10 at the SRO(001) surfaces are determined to be -6.13 and -8.57 eV (**Figure R14 b and f, Table R1**), suggesting a stable adsorption geometry between the Ru cluster and the substrate. We also observed significant electron transfer from the Ru cluster to the SRO substrate (**Table R1**), which is consistent with the Ru6 cluster and previous works (Hu et al. *Adv. Sci.* 2021, 8, 2001881). The adsorption of hydrogen at the reconstructed Ru/SRO surfaces was investigated by following the same procedure for Ru cluster. The preferred adsorption geometries for hydrogen adsorption are displayed for Ru1/SRO (**Figure R14 c top view, and Figure R14 c side view**), and Ru10/SRO (**Figure R14 g top view, and Figure R14 h side view**). The Gibbs free energy for hydrogen adsorption is 1.86 and 1.69 eV for Ru1/SRO and Ru10/SRO, respectively (**Table R1**), suggesting thermodynamically unfavorable for HER reactions. This is consistent with the previous report that Ru 10 has ground states with closed electronic shells, which is not suitable for molecular adsorption and surface catalysis (Dent et al. *PRB*, 1996, 54, 2191). In summary, we think that Ru6 is still the most possible cluster configuration although it's a challenge to confirm experimentally.

We summarized these results in the revised manuscript and hope the reviewer will be satisfied with our response.

Figure R14. Adsorption geometry of Ru 1 and Ru 10 clusters at the SRO (001) surface and the corresponding hydrogen adsorption configurations. Green, red, grey, and blue balls represent Sr, Ru, H, and Ru clusters, respectively. Ru cluster structures of a. Ru single atom, and b. Ru10 cluster. Adsorption geometry of b. Ru1/SRO, and f. Ru10/Ru. Top view of the hydrogen adsorption geometry for c. Ru1/SRO, and g. Ru10/SRO. d and h are the corresponding side view of hydrogen adsorption.

Ru cluster	Magnetic property	Adsorption energy (Ru/SRO, eV)	Electron transfer (Ru to SRO, e^-)	ΔG_{H^*} (eV)
Ru 1	Paramagnetic	-6.13	0.21	1.86
Ru 6	Ferromagnetic	-6.90	0.29	0.12
Ru 10	Ferromagnetic	-8.57	0.24	1.69

Table R1. Structure and properties information for different Ru clusters and Ru/SRO interface.

7. The DFT study uses a Ru6 cluster as catalysts, but why is the Ru6 used? Is there any experimental evidence that this is the case? From SEM there seems to be much larger particle formation. How much Ru is then available at the surface?

Response: To be honest, we agree with the reviewer that the experimental confirmation of the Ru cluster structure is a challenge, let alone they are formed *in situ* at the surface of the SRO bulk phase in a complicated electrochemical environment. However, we think **Ru6 is the most likely scenario because of the following reasons:**

a) Cox et al studied the magnetic properties of Ru_N clusters with *N* increased from 10 to 115 (Cox et al. *PRB*, 1994, 49, 12295). No magnetic deflection was observed for all of the investigated clusters within the experimental resolution at 60 K. However, clusters with *N* less than 10 have a magnetic moment of 0.32 μ_B/atom. This is consistent with our magnetic measurements, with the observation of a *weak ferromagnetic phase after surface reconstruction* (Fig 3j in the main manuscript).

b) Deng et al studied the electronic and magnetism properties of Ru_N clusters with *N* = 4, 6, 10, 13, 19, 43, and 55 (*PRB*, 1996, 54, 2191). They found that *Ru6 and Ru43 might have substantial reactivity towards H₂, N₂, and CO molecules catalysis*. However, Ru43 cluster is at the threshold to show bulklike properties of Ru metal.

c) Mori et al successfully obtained Ru clusters by the reduction of ruthenium salt (*ChemCatChem*, 2019, 11, 1963). Interatomic distance and coordination number obtained from the Extended X-ray Absorption Fine Structure analysis suggests the formation of Ru₆ geometry clusters. It looks like *Ru6 is one of the most thermodynamically stable phases among the clusters*.

From the above analysis, we think Ru₆ geometry could be the best choice for theoretical investigation.

As the reviewer point out, SEM images of the reconstructed surfaces suggested the formation of large particles. They are amorphous assemble of Ru clusters and partially SrO or other oxides, judging from SEM, XPS, and TEM measurements. This may suggest that *the in-situ reconstruction process is not homogeneous across the whole crystal surface*. Almost the same phenomena were reported by the in-situ activation of another family of layered compounds, delafossite oxides (Li et al, *ACS Energy Lett.*, 2019, 4, 2185; and Podjaski, et al. *Nat Catal.*, 2020, 3, 55). For the comparison of mass activities, we assume that all the materials in the reconstituted layer are pure Ru clusters (details can be seen in the response to question 1).

8. The authors indicate that “The relatively positive Ru atoms at the Ru1 site will weak the adsorption of the H reaction intermediate, promoting the Tafel step of hydrogen desorption, and finally resulting in the high intrinsic HER efficiencies.” Could the authors provide the reasoning behind this? This is contrary to what is being reported

for molecular catalysts, N-doped nanocarbons, and sulfur-rich MoS₂, where a slightly negative charge on the active site (due to doping) is beneficial for the HER since the active site will easily allow the electron transfer to the proton.

Response: We thank the reviewer for this interesting question. To answer the question of whether a positive charge or a negative charge accumulation is beneficial for the HER activities, we think it is necessary to check the original electronic structures of the investigated catalysts.

For carbon-based (graphene oxide) and two-dimensional (MoS₂) materials, the bottleneck for the high activities is the absence of electronics states for the formation of bondings with adsorbates (Figure R15, right side). Thus, the adsorption of hydrogen at thermodynamically stable surfaces is difficult. This explains the fact why these materials are located at the right side of the volcano plots with too positive Gibbs free energy (ΔG_{H^*}), such as 1.86 eV for graphene oxide (Yan et al. *Adv. Mater.*, 2018, 30, 1704156), 1.2 eV for NiPS₃ (Wang et al. *Adv. Funct. Mater.*, 2020, 1908708). 0.698 eV for 1T-MoS₂ (Huang et al. *Nat Commun.*, 2019, 10, 982), and 2.5 eV for 2H-MoS₂ (Li et al. *J Energy Chem.*, 2021, 62, 516). Thus, negative charge or charge accumulation on the active sites is necessary to strengthen the molecule's adsorption. This can be achieved by strategies such as doping, making defects, or breaking the local symmetry (strain engineering) (Xia et al. *J. Mater. Chem. A*, 2022,10, 19067).

Figure R14. Illustration of the relationship between charge transfer direction and HER activities.

On the contrary, the ΔG_{H^*} value for pure Ru is -0.41 eV, suggesting strong adsorption of hydrogen and unfavorable for the Tafel step of hydrogen desorption (Zhao et al. *ACS Catal.* 2020, 10, 11751) (**Figure R15, left side**). It has been well confirmed that the extraction of electrons from Ru is a good strategy to optimize the ΔG_{H^*} values. In this case, *the slight positive charge at the Ru site could weak the adsorption of the H reaction intermediate, promoting the Tafel step of hydrogen desorption*, and finally resulting in high intrinsic HER efficiencies. For example, the P atom could gain electrons from Ru and then decrease the ΔG_{H^*} value from -0.41 to -0.24 (Zhao et al. *ACS Catal.* 2020, 10, 11751). It was also confirmed that no matter which kind of substrates are chosen, electron transfer is from Ru to the carbon substrate (Hu et al. *Adv. Sci.* 2021, 8, 2001881). Here, we should admit that one cannot judge the trend of activity changes simply by charge transfer, d band theory is still one of the most decisive factors.

9. *The authors use a fairly complex equivalent circuit diagram during the EIS analysis, but there is no information regarding the meaning of it. The authors should provide further details. What is the origin of the two semicircles?*

Response: We are sorry for missing the information in understanding the EIS modeling. In the interpretation of the two semi-arc EIS data, an electrical equivalent circuit with two parallel time-constant model is generally used (**Figure R16 a**) (Jiang et al. *Angew. Chem. Int. Ed.* 2016, 55, 15240). However, the fitting is unsatisfactory with the above model. We find that *our impedance spectra can be fitted well with the two-semi arcs model (Figure R16 b) proposed by Lyons et al (J Electroanal. Chem., 2009, 631: 62)*. In the equivalent circuit, R_s represents the uncompensated solution resistance, C_{film} and R_{film} are related to the dielectric properties and the resistivity of the reconstructed Ru layer, while C_{dl} represents the double-layer capacitance. C_ϕ is the capacitance associated with the relaxation of the charge from the adsorbed reaction intermediate. R_p and R_e can be indexed to the kinetics of the interfacial charge transfer, with the former connected to the electroadsorption process and the latter representing the resistivity as a result of the dielectric interlayer. Such a two-charge

transfer process can be further confirmed by the analysis of Bode plots in the main manuscript.

Figure R16. Electrical equivalent circuits with a two-time constant model to simulate catalysts without (a) and with (b) heavy surface reconstruction.

We summarized the main results in the revised main manuscript and then provided the details in the supporting information.

Some other minor comments:

-There seems to be a mistake in the wire used to connect the catalysts. The experimental section indicates Cu, but in the “HER catalytic performance” section indicates Ti.

Response: We thank the reviewer for pointing out this mistake. We use Cu wire throughout this investigation (Ti wire was chosen occasionally in our previous works).

-The specific surface area of the SRO is mentioned in the main text, and in Supp information (Figure S5) but no value is given. This information should be specified.

Response: The electrochemical active surface area (ECSA) was determined by measuring the double-layer capacitance (C_{dl}) in the non-faradaic region since it is linearly proportional to ECSA. The ECSA of the activated SRO catalyst is determined to be 250 cm^2 from the double-layer capacitance (**Figure R5**).

- How much of the electrolyte resistance was corrected?

Response: We use the impedance spectra for resistance corrections. The solution resistance was determined to be 1.84Ω and 1.54Ω in $0.5 \text{ M H}_2\text{SO}_4$ and 1 M KOH at

room temperature, respectively, while the resistance of 0.95 Ω is obtained at 70 °C in 1 M KOH.

-Line 181 in the submitted manuscript, Fig. 2E is mentioned, but it should be 2F, when mentioning the 9000 mA/cm².

Response: We are sorry for this mistake. We have indexed the right figures and then checked the whole manuscript to avoid such mistakes.

-Indicate the methodology used to measure the bubble size.

Response: The Image J software was used to determine the bubble size. In each figure, ten bubbles were chosen manually, and then the average sizes of the bubbles were calculated by the software which is 132 and 78 μm for Pt and SRO, respectively.

-For the DFT computations. When evaluating the Gibbs free energy, the zero-point energy and entropy changes are required. But no information is provided about these quantities. Clearly indicate how these were evaluated.

Response: The Gibbs free energy of the adsorbed state H* (ΔG_{H^*}) can be calculated as:

$$\Delta G_{H^*} = \Delta E_H + \Delta E_{ZPE} - T\Delta S_H$$

where the ΔE_{ZPE} and ΔS_H are the difference in zero-point energy (ZPE) and the entropy between atomic hydrogen adsorption and hydrogen in the gas phase, respectively.

The zero-point energy ΔE_{ZPE} is obtained by

$$\Delta E_{ZPE} = E_{ZPE}^{nH^*} - E_{ZPE}^{(n-1)H^*} - \frac{1}{2}E_{ZPE}^{H_2}$$

where the $E_{ZPE}^{nH^*}$ and $E_{ZPE}^{(n-1)H^*}$ denote the ZPE of n and n-1 adsorbed atomic hydrogens on the catalyst without the contribution of the catalyst, respectively. For the $E_{ZPE}^{H_2}$, it is the ZPE of H₂ which is calculated to be 274.4 meV in this work. In our calculation, $E_{ZPE}^{H^*}$ is calculated to be 178.3 meV for Ru₂ site of Ru₆/SRO catalysts. Therefore, ΔE_{ZPE} is calculated to be 0.04 eV.

The entropy ΔS_H is obtained by

$$\Delta S_H \cong -\frac{1}{2} S_{H_2}$$

where the S_{H_2} is the entropy of H_2 gas under the standard condition.

We include these theoretical details in the supporting information of the revised manuscript.

-Given the value of the adsorption free energy, a near complete Hydrogen monolayer is expected, but only a single H atom is used during the evaluation. This can significantly affect the hydrogen-Ru interaction.

Response: To give an understanding of the effect of hydrogen coverage on the HER performance. We performed a new calculation by considering the full coverage of SRO/ Ru_6 surface. The adsorption model is relaxed to find the stable adsorption position as shown in **Figure R17**. As an indication of stable adsorption, the adsorption energy (ΔE_H) is calculated as below:

$$\Delta E_H = E_{nH^*} - E_{(n-1)H^*} - \frac{1}{2}E_{H_2}$$

where the E_{nH^*} and $E_{(n-1)H^*}$ represent the total energies of the catalyst with n and n-1 hydrogen atoms adsorption, respectively, while the E_{H_2} indicates the total energy of H_2 gas. The Gibbs free energy is calculated to be 1.8 eV. *The large positive adsorption energy may be caused by the ultra-small size of the Ru_6 cluster (only two Ru atoms are exposed), which leads to the strong repulsion of the adjacent H atoms.*

Figure R17. The energetically favorable configuration for Ru_6 /SRO catalysts that are covered by two H atoms.

REVIEWERS' COMMENTS

Reviewer #2 (Remarks to the Author):

The authors well addressed all the comments and revised the paper accordingly. The reviewer feels that the revisions are satisfactory and the paper is acceptable.

Reviewer #3 (Remarks to the Author):

In this revision, Guowei Li and co-authors have addressed all comments satisfactorily. They have performed additional experiments, simulations, and analysis to corroborate their claims. They have also modified the manuscript to include the new data and discussion. Therefore I have no additional comments.

Response to reviewers

Our point-to-point responses are as follows:

Reviewer #2 (Remarks to the Author):

The authors well addressed all the comments and revised the paper accordingly. The reviewer feels that the revisions are satisfactory and the paper is acceptable.

Response: We thank the referee for the careful reading of our manuscript and for providing useful comments.

Reviewer #3 (Remarks to the Author):

In this revision, Guowei Li and co-authors have addressed all comments satisfactorily. They have performed additional experiments, simulations, and analysis to corroborate their claims. They have also modified the manuscript to include the new data and discussion. Therefore I have no additional comments.

Response: We deeply appreciate the referee for his/her valuable comments, which are important to improve this work.